# Snow depth variability in the Northern Hemisphere mountains observed from space

Hans Lievens [1,2]*, Matthias Demuzere [2,3], Hans-Peter Marshall[4,5], Rolf H. Reichle [6], Ludovic Brucker [6,7], Isis Brangers[1], Patricia de Rosnay [8], Marie Dumont [9], Manuela Girotto [6,7,10], Walter W. Immerzeel [11], Tobias Jonas[12], Edward J. Kim[6], Inka Koch [13], Christoph Marty [12], Tuomo Saloranta[14], Johannes Schöber [15] & Gabrielle J.M. De Lannoy[1]

Accurate snow depth observations are critical to assess water resources. More than a billion people rely on water from snow, most of which originates in the Northern Hemisphere mountain ranges. Yet, remote sensing observations of mountain snow depth are still lacking at the large scale. Here, we show the ability of Sentinel-1 to map snow depth in the Northern Hemisphere mountains at 1 km² resolution using an empirical change detection approach. An evaluation with measurements from ~4000 sites and reanalysis data demonstrates that the Sentinel-1 retrievals capture the spatial variability between and within mountain ranges, as well as their inter-annual differences. This is showcased with the contrasting snow depths between 2017 and 2018 in the US Sierra Nevada and European Alps. With Sentinel-1 continuity ensured until 2030 and likely beyond, these findings lay a foundation for quantifying the long-term vulnerability of mountain snow-water resources to climate change.

[1] Department of Earth and Environmental Sciences, KU Leuven, Heverlee, Belgium. [2] Laboratory of Hydrology and Water Management, Ghent University, Ghent, Belgium. [3] Department of Geography, Ruhr-University Bochum, Bochum, Germany. [4] Department of Geosciences, Boise State University, Boise, ID, USA. [5] Remote Sensing and GIS Center, U.S. Army Cold Regions Research and Engineering Laboratory, Hanover, NH, USA. [6] NASA Goddard Space Flight Center, Greenbelt, MD, USA. [7] GESTAR, Universities Space Research Association, Columbia, MD, USA. [8] European Centre for Medium-Range Weather Forecasts, Reading, UK. [9] Université Grenoble Alpes, Université de Toulouse, Météo-France, Grenoble, France, CNRS, CNRM, Centre d'Etudes de la Neige, Grenoble, France. [10] Environmental Science and Policy Management Department, University of California, Berkeley, CA, USA. [11] Department of Physical Geography, Utrecht University, Utrecht, The Netherlands. [12] WSL Institute for Snow and Avalanche Research SLF, Davos, Switzerland. [13] International Centre for Integrated Mountain Development, Kathmandu, Nepal. [14] Hydrology Department, Norwegian Water Resources and Energy Directorate NVE, Oslo, Norway. [15] TIWAG, Tiroler Wasserkraft AG, Innsbruck, Austria. *email: hans.lievens@kuleuven.be

Snow has a large-scale cooling effect on our planet by reflecting most of the incoming solar radiation and by dissipating energy during seasonal melt[1,2]. Water resources from snowmelt, of which the majority originates in mountain ranges, are indispensable. For instance, they provide drinking water to over a billion people[3,4], supply 3/4th of the crop production in the western US[5], generate hydro-electric power worldwide[4], and sustain the vast urban settlements in Himalayan watersheds[6–8]. Despite this importance, we lack quantitative estimates of how much snow is stored in each mountain range on Earth[9]. The current estimates from the interpolation of local measurements are unrealistic where measurements are sparse[9], whereas estimates from numerical weather prediction systems are poor due to the large uncertainty in mountain snowfall[4].

Remote sensing provides essential snow depth information, yet currently operational observations have shortcomings[9]. Passive microwave observations[10,11] typically exclude mountains because their coarse (~25 km) footprints cannot resolve the spatial variability, and the observations saturate in deep snow (>0.8 m depth)[12,13]. Airborne lidar systems[5,14] are likely the most accurate to date, but are limited to targeted mountain areas and favorable weather conditions. Some alternative methods have shown promise at the local scale. These include structure-from-motion[15,16], constructing snow depth from overlapping image pairs, X-/Ku-band (8–18 GHz) Synthetic Aperture Radar (SAR)[17–19], retrieving snow properties based on the backscattered energy from an illuminated snowpack, and SAR interferometry[20,21], measuring changes in the radar signal phase from refraction in the snow. A community effort to compare some of the above-mentioned and more experimental remote sensing techniques is currently ongoing within the NASA SnowEx activity that should ultimately lead to a space mission designed for snow[22]. New and robust satellite observations are critically needed to fill the mountain-snow observation gap[4].

In this study, we demonstrate the ability of the ESA and Copernicus Sentinel-1 constellation (two satellites: 1A and 1B) to map snow depth across the Northern Hemisphere mountains. Presently, Sentinel-1 is the only SAR mission providing high-resolution backscatter measurements (at C-band; 5.4 GHz) with a revisit time of 6 days suitable for snow monitoring. Given the strong absorption of C-band microwave radiation by wet snow, monitoring applications have so far focused on the mapping of melt[23]. The use of C-band backscatter for estimating snow depth (or mass, related to depth by the density) has long been swept aside after early satellite measurements had shown a limited sensitivity[24,25]. However, these studies were mostly surveying shallow snow outside mountain environments and, more importantly, were limited to backscatter measurements in co-polarization. Cross-polarized backscatter measurements were to-date only investigated at the local scale using tower installations, with strongly contradicting results[26,27]. Here, we demonstrate the value of including cross-polarized backscatter measurements from C-band satellite to retrieve snow depth in mountainous areas at the large scale.

The Sentinel-1 snow depth retrievals over the Northern Hemisphere mountain ranges allow for near-real-time monitoring that complements current snow tracking systems from the World Meteorological Organization (WMO). Unlike currently available space-borne passive microwave measurements, Sentinel-1 measurements are suitable for retrieving snow depth in mountainous environments owing to their high resolution and the herein demonstrated sensitivity to deep snow. We provide clear evidence of the value of the empirical Sentinel-1 retrievals through a comparison against point-scale snow depth measurements at ~4000 sites and coarse-scale global reanalysis data. Over the US Sierra Nevada and the European Alps, snow depths of two

consecutive winters are stratified by elevation to highlight inter-annual differences. While resembling the large-scale patterns of the reanalysis data, Sentinel-1 more accurately reveals the spatial detail and the elevation profile of snow depth. Finally, we estimate total snow volumes for the top 100 snowiest mountains in the Northern Hemisphere to support their water resources monitoring.

## Results

**Sentinel-1 backscatter signatures over snow.** The Sentinel-1 constellation routinely illuminates the land surface with C-band radiation and measures the backscatter ($\sigma^0$) in co-polarization and cross-polarization, i.e., vertical-vertical (vv) and vertical-horizontal (vh) transmit-receive, respectively. The total backscatter ($\sigma_{pq}^{tot}$) for transmit-receive polarizations $p$ and $q$, originating from a snowpack without vegetation, can be approximated by a four-component model[28]:

$$\sigma_{pq}^{tot} = \sigma_{pq}^{air-snow} + \sigma_{pq}^{snowvol} + \sigma_{pq}^{snowvol-grnd} + e^{\left(-2\tau_p/\cos\theta\right)} \cdot \sigma_{pq}^{grnd}$$

(1)

with $\sigma_{pq}^{air-snow}$ the backscatter from the air-snow interface, $\sigma_{pq}^{snowvol}$ the backscatter from the snow volume, $\sigma_{pq}^{snowvol-grnd}$ the higher-order interactions between the snow volume and the ground, and $\sigma_{pq}^{grnd}$ the backscatter from the ground, attenuated by the snowpack through $e^{\left(-2\tau_p/\cos\theta\right)}$. The amount of attenuation depends on the optical thickness of the snow ($\tau_p$) and the radar incidence angle ($\theta$). Commonly, $\sigma_{pq}^{air-snow}$ and $\sigma_{pq}^{snowvol-grnd}$ are neglected[29] and thus not further considered here.

Sentinel-1 $\sigma^0$ measurements are processed at 1 km² resolution over the Northern Hemisphere mountains[30] (Fig. 1a), using standard processing techniques (Fig. 2 and Methods), and their temporal signatures are compared to those of in situ snow depth measurements. Figure 3 illustrates the temporal evolution in $\sigma^0$ and snow depth measurements at 4 representative sites, along with corresponding snow presence observations at 1 km² resolution from the Interactive Multi-sensor Snow and Ice Mapping System (IMS[31]). The co-polarized $\sigma_{vv}^0$ measurements (Fig. 3, left column) show little variation throughout the winter, due to the limited absorption or scattering by dry snow in vv-polarization at C-band. In contrast, a sharp (~5 dB) decrease in $\sigma_{vv}^0$ is observed with the wetting and depletion of the snow through spring. This confirms previous studies that demonstrated the limited sensitivity of C-band co-polarized satellite measurements to changes in dry snow and the large absorption and reflection of the signal by wet snow[23–25]. It also confirms previous findings that C-band vv-polarized measurements are suitable for snowmelt mapping[23]. The increase in $\sigma_{vv}^0$ towards the end of the snowmelt period in late spring is caused by the decreasing surface area covered by wet snow that contributes to absorption[32]. Simultaneously, surface thaw and vegetation green-up in snow-free areas can increase scattering[32]. The evolution of $\sigma_{vv}^0$ during the snow season is dominated by the fourth term in Eq. (1): the backscatter from the ground surface, which is attenuated by the snowpack only when the snow is wet. The second term, representing the scattering in vv-polarization within a dry snow volume, is generally too small to have a noticeable impact at C-band.

The Sentinel-1 cross-polarized $\sigma_{vh}^0$ measurements (Fig. 3, middle column) gradually increase with the accumulation of (dry) snow during the winter. This suggests an increasing depolarization of the incoming v-polarized signal through anisotropic or multiple scattering on ice crystals, bonds or clusters of ice crystals, or inhomogeneities within the snow volume[17,19,26,33–35]. As the thickness of the snowpack increases,

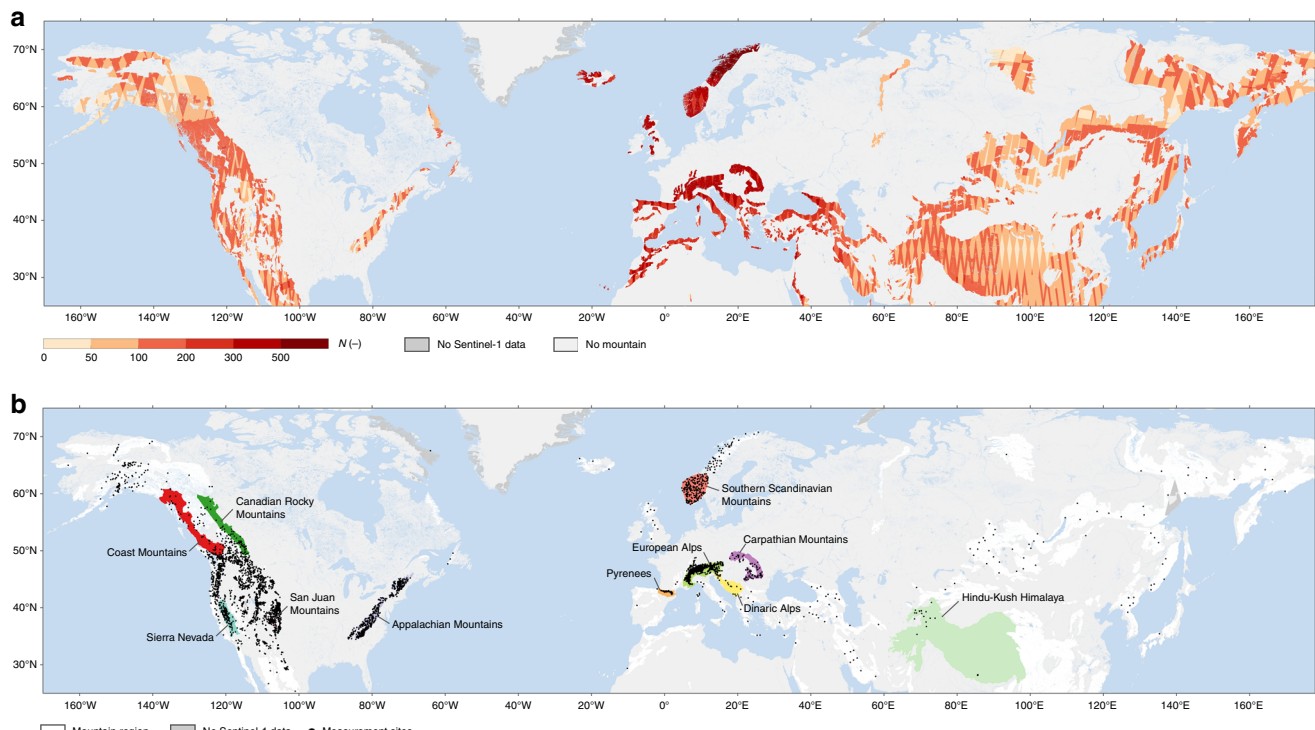

**Fig. 1** Study domain and data. **a** The study domain corresponds to the Northern Hemisphere mountains, north of 20°N. Colors indicate the number (*N*; dimensionless) of Sentinel-1 backscatter measurements for the period September 2016 through August 2018. *N* is calculated after combining the ascending and descending Sentinel-1A and Sentinel-1B measurements at the daily timescale, and is limited to mountain areas observed by Sentinel-1 in vv-polarization and vh-polarization. Measurements are on average available every ~2 days in Europe and every ~4 days to ~2 weeks in North America and Asia. **b** In situ snow depth measurement sites within the study domain. The sources of the in situ snow depth measurements are given in Table 1. The majority of the 4175 sites is located in the western US and Europe. Measurements in Asia are more scarce. Colors highlight 11 focus mountain ranges, which were selected based on the availability of in situ measurements and based on differences in geographic location, climate, distance to the coast, elevation, and ruggedness, i.e., representing a broad variety of snow conditions

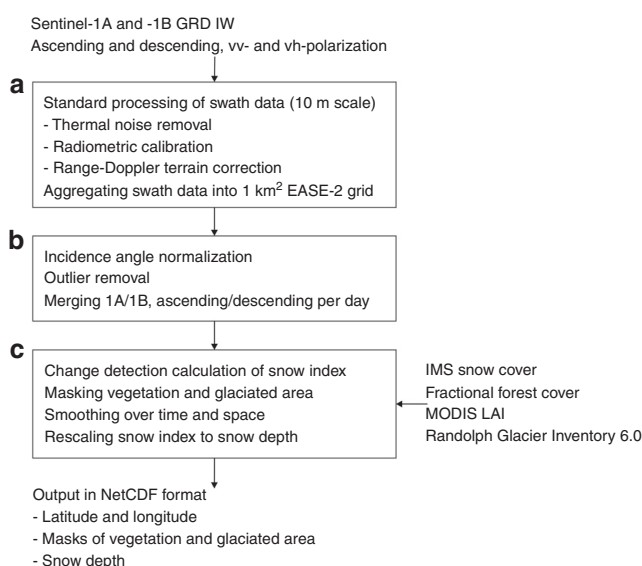

**Fig. 2** Flowchart of the Sentinel-1 processing. **a** The standard backscatter processing techniques are performed using the Google Earth Engine Python's api. **b** Further processing steps, and **c** the application of the retrieval algorithm are performed offline

the path length of the radar signal through the snow increases, increasing the opportunities for scattering[19]. Note that this anisotropic and multiple scattering in theory also impacts the backscatter in co-polarization ($\sigma_{vv}^0$), but with a negligible

contribution compared to that of the ground surface. During snow ablation, $\sigma_{vh}^0$ decreases considerably, which we hypothesize is caused by the absorption and reflection of the signal by wet snow, and a decreasing amount of snow volume scattering in a shallowing snowpack. To explain the variations in $\sigma_{vh}^0$, both the second and fourth terms in Eq. (1) are needed. The snow volume scattering component in vh-polarization is no longer negligible compared to the contribution of the ground surface.

The cross-polarization ratio $\sigma_{vh}^0/\sigma_{vv}^0$ (in linear scale, converted to dB) shows overall a stronger correlation with snow depth than $\sigma_{vh}^0$ (Fig. 3, middle and right columns). Taking the ratio may partially eliminate the effects of temporal changes in the ground surface, vegetation, or snow conditions, which similarly impact both co- and cross-polarization. In autumn and winter, $\sigma_{vh}^0/\sigma_{vv}^0$ increases due to the increased amount of volume scattering $\sigma_{vh}^0$ for approximately constant surface scattering $\sigma_{vv}^0$. During spring melt, the ratio decreases because of the relatively higher decrease of $\sigma_{vh}^0$ compared to that of $\sigma_{vv}^0$. Changes in snow properties between dry and wet conditions, such as in snow microstructure and liquid water content, potentially modify the proportionality of volume versus surface scattering, and therefore the sensitivity of the ratio to snow depth. Thereby, the sensitivity to snow depth for wet snow is much more uncertain. Strong melt events, with associated high liquid water contents, may cause fluctuations in $\sigma_{vh}^0/\sigma_{vv}^0$ as the signal is more strongly reflected and absorbed by wet snow layers[27]. Although this effect may be less severe at C-band than at higher frequencies and may occasionally be alleviated by the refreezing of snow[27] prior to the early-morning (6 a.m.) and evening (6 p.m.) overpass times of Sentinel-1, it is likely to have a

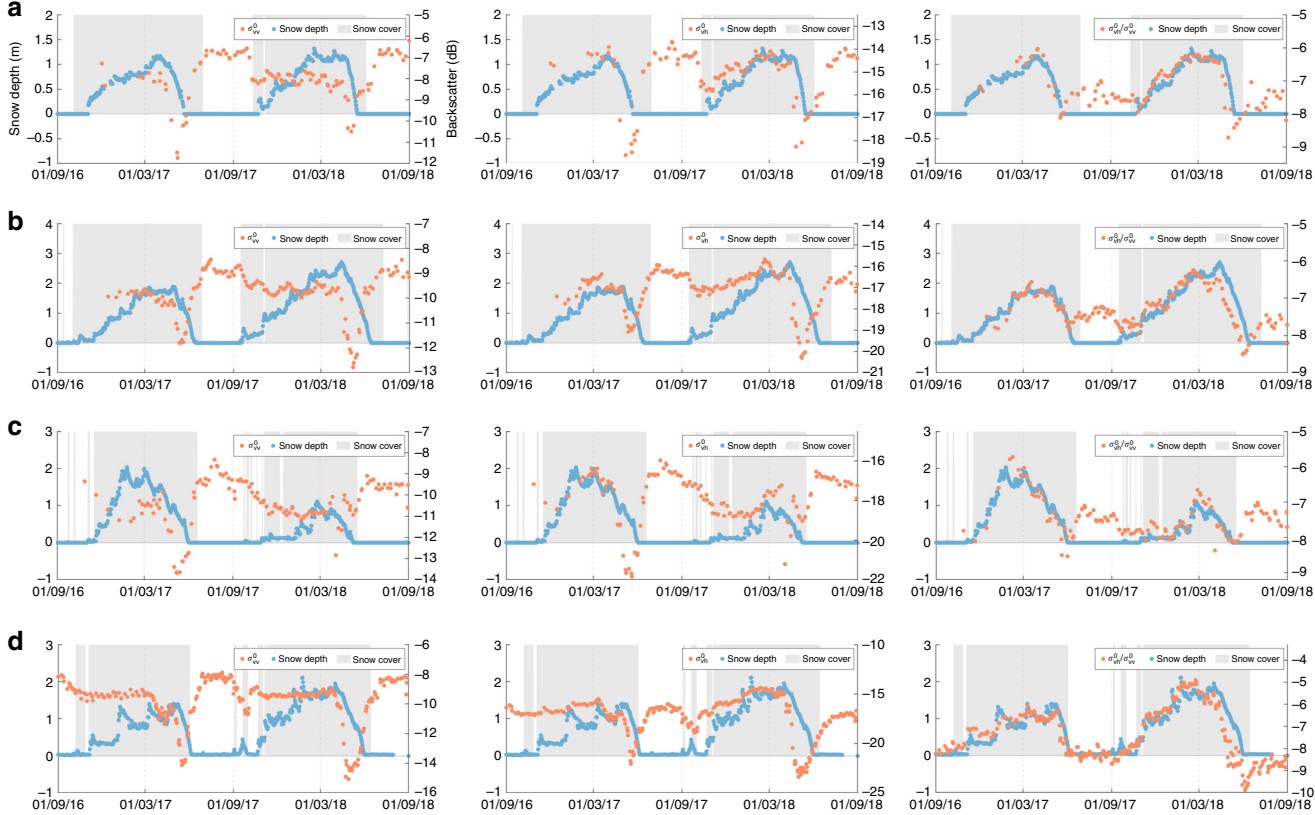

**Fig. 3** Backscatter and snow depth time series at 4 sites. The left column shows the co-polarized backscatter ($\sigma^0_{vv}$; in dB), the middle column the cross-polarized backscatter ($\sigma^0_{vh}$; in dB), and the right column the cross-polarization ratio ($\sigma^0_{vh}/\sigma^0_{vv}$; in dB). Snow cover presence (dimensionless) from IMS is shown in gray. Site details, including mountain range and latitude/longitude/elevation: **a** Omineca Mountains, British Columbia, Canada (57.53°N/−126.73°E/1331 m), **b** Absaroka Range, Wyoming, USA (45.22°N/−110.24°E/2697 m), **c** San Juan Mountains, Colorado, USA (37.71°N/−107.51°E/3535 m), **d** European Alps, Austria (46.87°N/10.71°E/2464 m). The ranges of the backscatter (dB) and snow depth (m) measurements have been adjusted between sites to improve visualization

confounding impact on the C-band sensitivity to snow depth. Similarly, the backscatter signals are impacted by successive melt-refreeze cycles that can modify the microstructure and stratigraphy of the snowpack. The primary objective of this study is to map the snow depth for dry snow conditions. However, we do not exclude wet snow conditions from the analysis and assess the performance of the retrievals throughout the snow season.

**Sentinel-1 snow depth retrievals**. The retrievals of ~weekly snow depth at 1 km² resolution for September 2016 through August 2018 over the Northern Hemisphere mountains are based on the temporal changes in the Sentinel-1 backscatter polarization ratio ($\sigma^0_{vh}/\sigma^0_{vv}$) and scaled to the range of snow depth measurements at in situ sites (Fig. 2), as discussed in the Methods section. The temporal and spatial resolutions of the ~weekly 1 km² retrievals meet the requirements for watershed-scale applications in mountain regions[4], and with the Sentinel-1 follow-on missions (1C and 1D) multi-decadal data will be provided for trend analysis. Figure 4 illustrates the Sentinel-1 snow depth averaged over February 2018. We select the month of February as it corresponds with the climatological maximum snow-cover extent according to the WMO Global Cryosphere Watch. The insets illustrate details in snow depth variability in the western United States and Hindu-Kush Himalaya (Fig. 4b, c).

An assessment based on measurements at 4175 locations (Fig. 1b, Table 1 and Methods) reveals the effectiveness of the Sentinel-1 retrievals to capture the overall spatial snow depth

variability. For February 2018, the spatial correlation ($R_s$) with monthly-averaged snow depths from all in situ measurement sites is 0.76 (other months in Table 2). Both the variability between and within mountain ranges is accurately captured. For instance, when averaging snow depths for 11 selected mountain ranges (Fig. 1b), the $R_s$ between these ranges is 0.92. Within these mountain ranges, the $R_s$ is 0.72 on average. The latter value is lower due to the limited number of sites in some ranges and the potentially large differences in spatial representativeness. More specifically, measurement sites suitable for instrumentation are typically in relatively flat areas and may not represent the snow conditions of the surrounding slopes[36] that are included in the satellite footprint. The spatial variability in snow depth characterized by Sentinel-1 C-band SAR data is a clear improvement over currently available global estimates from reanalysis data, such as MERRA-2 (Modern-Era Retrospective analysis for Research and Applications, Version 2[37]; selected because it does not assimilate snow depth measurements and thus provides an independent comparison). Corresponding $R_s$ values for MERRA-2 in February 2018 are 0.36, 0.67, and 0.29 for the overall, in-between and within mountain-range variability, respectively.

Figure 5 displays the weekly evolution in snow depth over two consecutive winters averaged over all Northern Hemisphere mountain ranges and over the 11 focus ranges. Across the Northern Hemisphere (Fig. 5a), the retrievals accurately characterize the accumulation of (dry) snow, whereas a slight underestimation is noticeable during snow ablation from March onwards. This underestimation is likely caused by wet snow

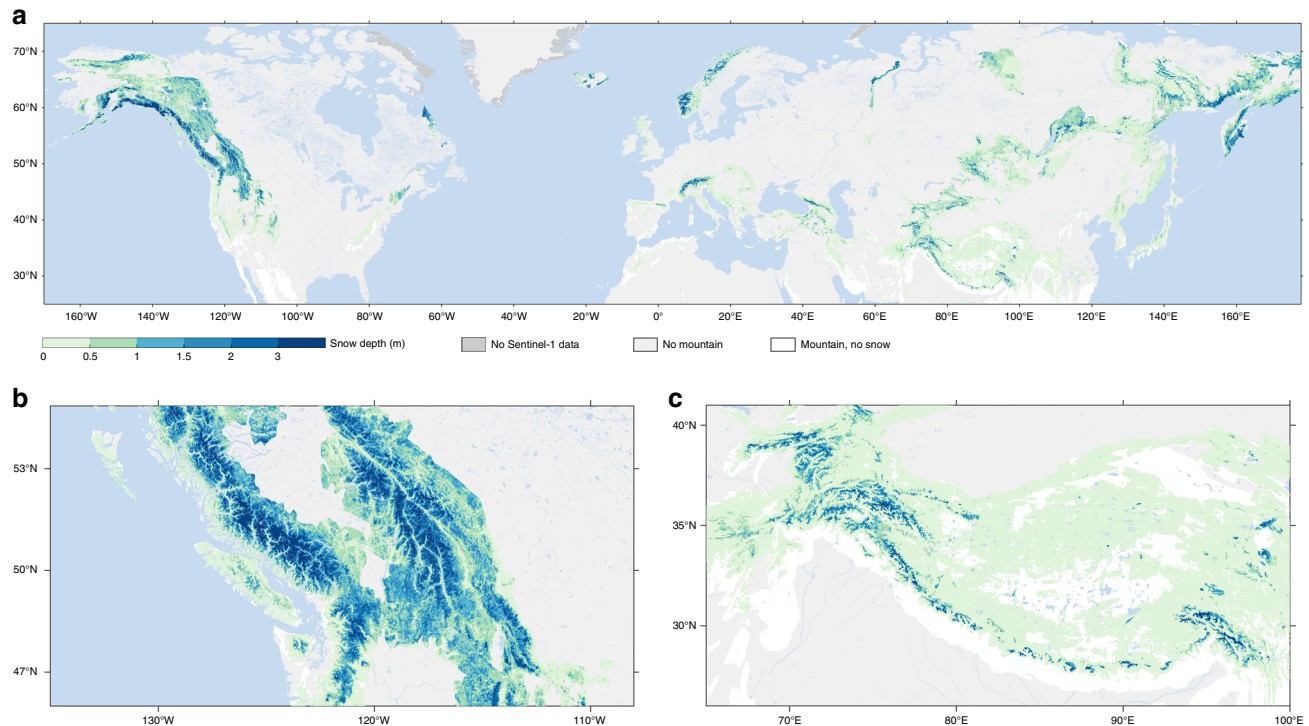

**Fig. 4** Monthly snow depth retrieved by Sentinel-1. **a** Snow depth (m) over the Northern Hemisphere mountains. **b** Enlargement of a part of the western US and Canada. **c** As in panel **b**, but for a part of the Hindu-Kush Himalaya region. The ~weekly 1 km² resolution Sentinel-1 retrievals are averaged over the month of February, 2018

**Table 1 Description of the in situ measurements**

| Region | Network name | Provider | N sites | Median elevation | References |
|---|---|---|---|---|---|
| Global | SYNOP | ECMWF | 894 | 432 | 51 |
| Global | GHCN-D | NOAA | 3303 | 1024 | 59,60 |
| USA | SNOTEL | USDA-NRCS | 733 | 2409 | – |
| USA | SNOLITE | USDA-NRCS | 16 | 1982 | – |
| USA & Canada | Snow course/aerial marker | USDA-NRCS | 615 | 1981 | – |
| USA & Canada | Cooperator snow sensors | USDA-NRCS | 32 | 1591 | – |
| Canada | ASWS | British Columbia Government | 50 | 1510 | – |
| Nepal | – | ICIMOD, DHM, KU, Utrecht University, NVE Norway | 9 | 4888 | 61–63 |
| France | – | Météo France | 129 | 1713 | – |
| Austria | – | ZAMG | 209 | 864 | – |
| Austria | – | TIWAG | 9 | 1920 | 64,65 |
| Switzerland | BEOB, IMIS | WSL SLF | 407 | 1560 | 66 |

Details of the global and regional in situ networks, including the regional coverage, network name (if applicable), data provider, number (N; dimensionless) of sites, median elevation (m), and references (if applicable). The number of sites is reported before averaging into 1 km² EASE-2 grid cells and the total thus exceeds 4175

**Table 2 Correlations between measured and retrieved snow depths**

| Month | Overall $R_s$ (−) | $R_s$ (−) between mountain ranges | $R_s$ (−) within mountain ranges |
|---|---|---|---|
| December | 0.67 (<0.01) | 0.96 (<0.01) | 0.58 (<0.01) |
| January | 0.72 (<0.01) | 0.93 (<0.01) | 0.60 (0.02) |
| February | 0.76 (<0.01) | 0.92 (<0.01) | 0.72 (<0.01) |
| March | 0.69 (<0.01) | 0.85 (<0.01) | 0.67 (<0.01) |
| April | 0.72 (<0.01) | 0.84 (<0.01) | 0.60 (0.04) |

Spatial correlations ($R_s$; dimensionless) and their statistical significance level (p; in parentheses) are shown for monthly-averaged Sentinel−1 and in situ snow depth from December 2017 through April 2018. To calculate the overall $R_s$, all available measurements across the Northern Hemisphere are used. The $R_s$ between mountain ranges is calculated from snow depths averaged per mountain range for the 11 ranges shown in Fig. 1b. The $R_s$ within mountain ranges is the average of the spatial correlations within each of these ranges

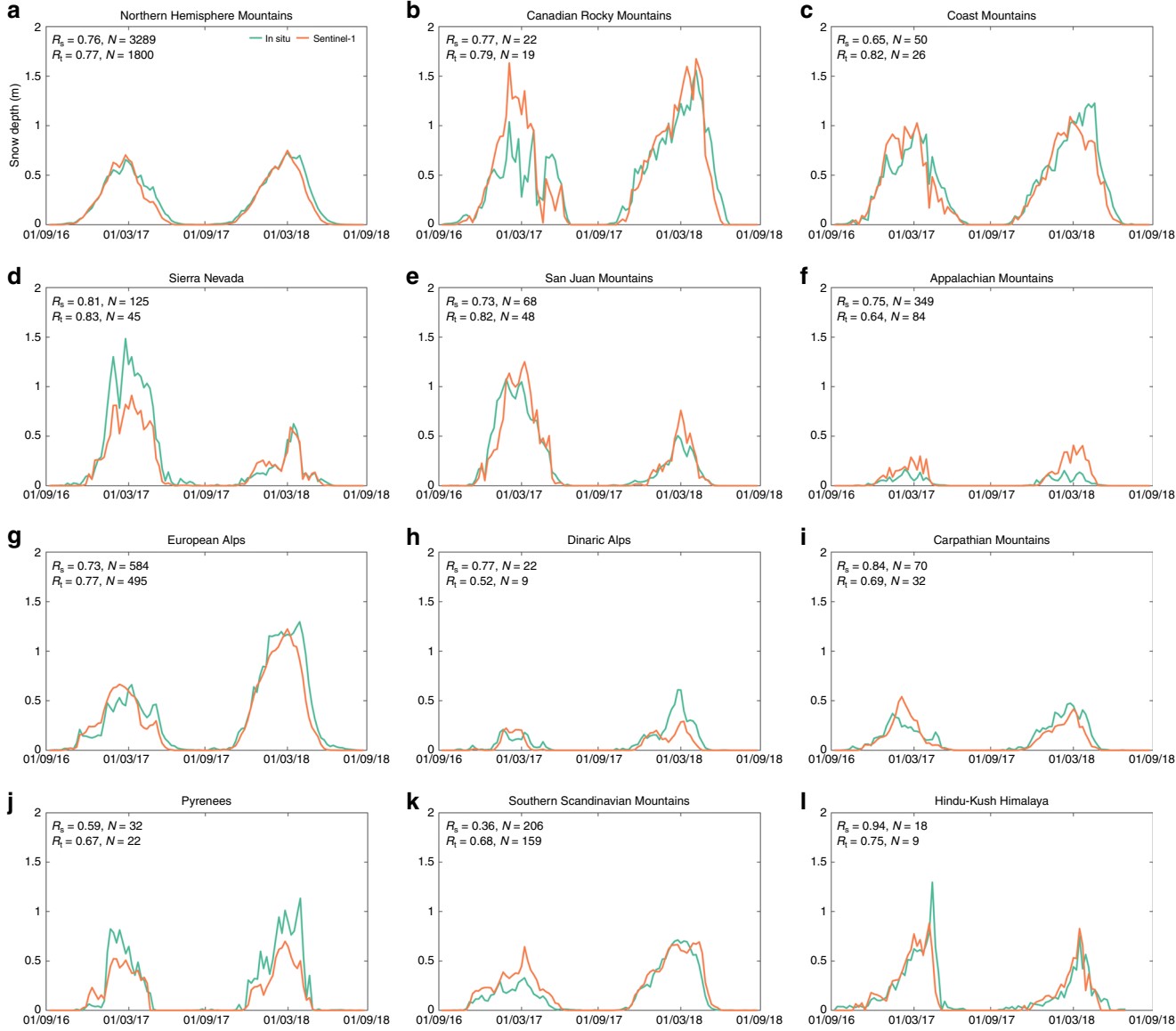

**Fig. 5** Time series of snow depth measurements and retrievals. **a** The average weekly snow depth (m) over all measurement sites and corresponding Sentinel-1 grid cells across the Northern Hemisphere. **b–l** As in a, but averaged over the focus mountain ranges (Fig. 1b). The performance metrics shown in the top left of each figure include the temporal correlation ($R_t$; dimensionless), the February spatial correlation ($R_s$; dimensionless), and the number of sites ($N$; dimensionless) used to calculate the metrics

conditions partly reflecting and absorbing the radar signal. However, the underestimation in spring does not occur in all mountain ranges. For example, the temporal evolution in the US Sierra Nevada (Fig. 5d) during 2017–2018, including the spring melt phase, is accurate. Similarly, no underestimation is noticeable in the San Juan Mountains (Colorado, US; Fig. 5e) and Southern Scandinavian mountains (Norway; Fig. 5k). Future research shall therefore investigate under which precise conditions wet snow could cause underestimation. In some ranges, such as the Canadian Rocky Mountains (Fig. 5b) and US Sierra Nevada (Fig. 5d), the evolution of snow depth in the winter of 2016–2017 is less accurate than in 2017–2018, due to the smaller number of Sentinel-1 observations.

Figure 6 shows histograms of performance metrics that evaluate the correspondence between time series of weekly snow depth measurements and Sentinel-1 retrievals across the Northern Hemisphere mountain area, from September 2016 through August 2018. Temporal correlation coefficients ($R_t$; gray histogram in Fig. 6a) most frequently range between 0.8 and 0.9, have a mean of 0.77 (with significance level $p < 0.01$), and are distributed with a negative skew. When zero snow depths are excluded from the analysis (white histogram), the associated reduction in the snow depth range decreases the mean correlation ($R_t = 0.65$; $p < 0.01$). The lowest temporal correlations occur in sites with shallow snow depth and short or intermittent snow cover. The mean-absolute-error (MAE) histogram (Fig. 6b) has a positive skew, with values most frequently below 0.1 m and a mean of 0.18 m. When we exclude zero values from the analysis, the average MAE increases to 0.31 m. The bias (Fig. 6c) remains within ± 0.1 m for most of the sites, has a mean of −0.01 m and a nearly normal distribution. Large errors (MAE and bias) can again originate from the differences in representativeness between point-scale measurements and grid-scale retrievals.

For reference, the most accurate snow depth estimates today are likely provided by airborne lidar, offering decimeter-scale accuracy in surface height (either snow-on or snow-off) over local

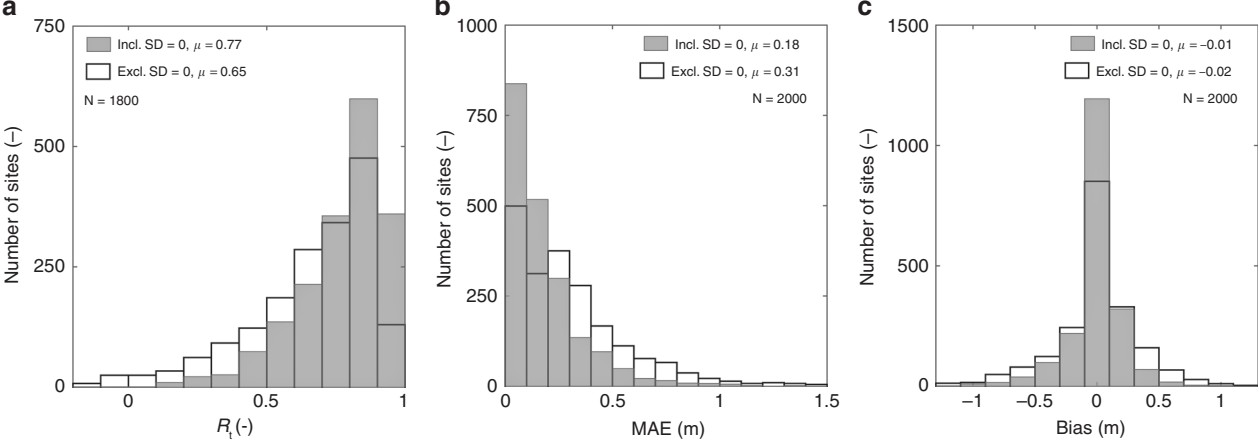

**Fig. 6** Histograms of retrieval performance metrics. **a–c** Histograms of temporal correlation ($R_t$; dimensionless), mean-absolute-error (MAE; m), and mean of retrievals minus measurements (bias; m), calculated from time series of weekly snow depth measurements and corresponding Sentinel-1 retrievals, for September 2016 through August 2018. Gray histograms are calculated with all reported measurements, including zero values (Incl. SD = 0); white histograms exclude zero values (Excl. SD = 0), with SD the snow depth (m). Metrics indicate the mean value ($\mu$) and the number of sites ($N$; dimensionless)

areas[14]. As an example, lidar snow depth estimates from the Airborne Snow Observatory have an MAE of ~0.08 m (at the 15 m × 15 m scale) with respect to manual in situ measurements over a relatively flat area near Tioga Pass in the Sierra Nevada, California, USA[5]. The Advanced Microwave Scanning Radiometer-Earth Observing System (AMSR-E) snow depth retrievals, available at global-scale, have correlations of ~0.25 and ~0.41 with estimates from the Snow Data Assimilation System (SNODAS) and WMO in situ measurements[12], and an MAE of 0.20 m against WMO in situ measurements[10]. However, a direct comparison with the Sentinel-1 retrieval performance is hampered by the limitation of the AMSR-E retrievals to areas without complex topography and with snow depths typically below 0.8 m.

**Inter-annual differences stratified by elevation.** Characterizing the inter-annual change in snow depth and how it varies with elevation is critical to assess the vulnerability of mountain systems to climate change. For snow cover, there is considerable evidence that global warming reduces the area and duration, thereby reinforcing climate change[38]. Recent studies indicate that this impact may be stratified by elevation, with projections of decreased snow cover at low or medium elevation, but no clear trend at high elevation[4,13,39,40]. Whether snow depth (and mass) are similarly affected remains uncertain, particularly in mountain areas where adequate observations are missing[4,9]. The most accurate trend estimates of mountain snow depth are currently provided by model reanalysis, although with only moderate consistency between different reanalysis products[41,42]. Unfortunately, within the complex topography of a mountain range, reanalysis data are too coarse for a stratification by elevation to be meaningful.

Figure 7 illustrates the benefit of Sentinel-1 over the European Alps and the US Sierra Nevada for two consecutive years with contrasting snow conditions. In the European Alps, snow depth in February 2018 greatly exceeded that of February 2017. The Alps were hit by several episodes of extreme snowfall in January 2018, caused by a low-pressure area over the western Mediterranean that brought moist air northwards and resulted in the anomalously high snow depths. The Sierra Nevada featured exceptionally deep snow in February 2017, caused by a series of atmospheric river events[43,44], whereas the snow depth in

February 2018 was relatively low. Over both the Alps and Sierra Nevada, similar large-scale patterns in snow depth differences (February 2018 minus 2017) are seen in the Sentinel-1 retrievals (Fig. 7a, d) and MERRA-2 reanalysis data (Fig. 7b, e). But the 1 km² Sentinel-1 retrievals provide a much more detailed representation, showing a gradual increase with elevation that closely follows that of the measurements and is poorly represented in MERRA-2 (Fig. 7c, f). Averaged over the 11 selected mountain ranges, the $R_s$ between snow depth measurements and elevation is 0.58. A similar $R_s$ is observed for the Sentinel-1 retrievals (0.50), whereas the MERRA-2 data are overall much less constrained by elevation ($R_s$ of 0.12).

**Mountain range snow volumes.** Worldwide, mountain snow-packs seasonally store and release substantial amounts of water[45]. Observations of the total snow volume (depth × area) in a mountain range can be used to estimate this water storage as a mass, if snow density is known. At the global scale, snow density can for instance be estimated from information on the snow climate class and mechanical compaction, depending on snow depth and day of the year[46,47]. Figure 8 shows the snow volume (km³) retrievals for February 2018 from Sentinel-1 for the top 100 snowiest mountain ranges in the Northern Hemisphere, excluding glaciated areas[48]. The largest snow volume (~380 km³) by far is in the Coast Mountains in western Canada, featuring a deep snowpack over a vast area. Given the maritime snow climate of the Coast Mountains, the February snow density is ~310 kg m⁻³. Correspondingly, the first-order estimate of the total water mass in this mountain range is ~11.8 × 10¹⁰ ton (or equally, ~400 kg m⁻²). Averaging the relatively few (48) measurements in this area results in a volume of 260 km³ (black cross in Fig. 8), considerably below the 380 km³ area-wide snow volume retrieval from Sentinel-1 (bar). This significantly lower estimate is likely due to the locations of the in situ measurements. About 75% of the sites are located in the southern half of the mountain range in a warmer climate. Moreover, 70% are located below the mean elevation (964 m) of the mountain range, with the highest site (1835 m) much below the highest mountain peak (4019 m). Confidence in the area-wide Sentinel-1 snow volume retrievals is supported by the close agreement between the cross-masked Sentinel-1 retrievals (i.e., averaged only over grid cells that include measurement sites; purple dots in Fig. 8) and the measurements (black crosses).

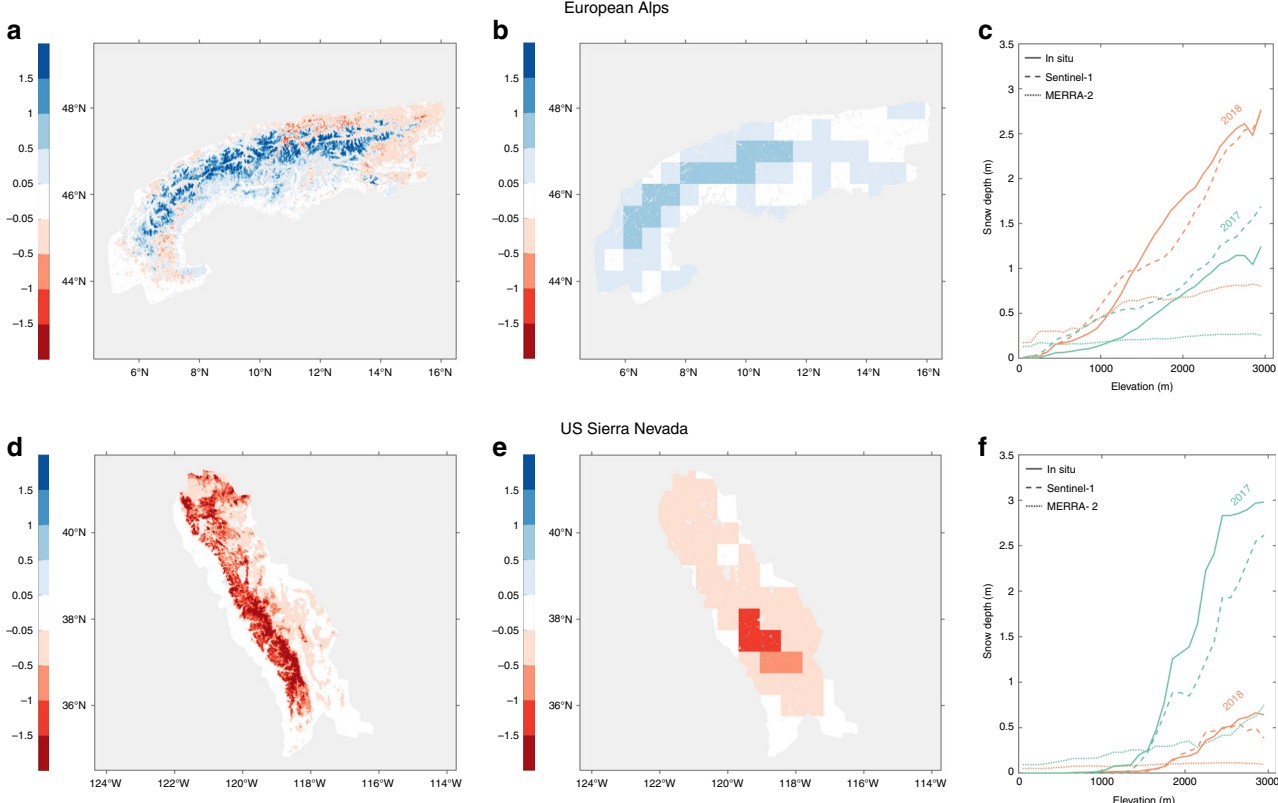

**Fig. 7** Inter-annual snow depth differences stratified by elevation. **a** Snow depth differences (February 2018 minus February 2017; in m) according to 1 km² Sentinel-1 retrievals over the European Alps. **b** As in a, but for coarse-resolution (0.625° × 0.5°) MERRA-2 reanalysis data. **c** Snow depth (m) of February 2017 and February 2018 stratified by elevation (m), for in situ measurements, Sentinel-1 retrievals and MERRA-2 reanalysis data (subsampled to the 1 km² grid). **d–f** As in panels **a–c**, but for the US Sierra Nevada mountain range. Source data for c,f are provided as a Source Data file

For most of the other featured mountain ranges in North America, snow volumes compare reasonably well between area-wide and cross-masked Sentinel-1 retrievals and in situ measurements. Mountain ranges with noticeable differences are mostly limited to those with few measurement sites, such as the Mackenzie and Chugach Mountains and the Alaska Range. Similar findings apply to the European ranges, whereas more contrasting results are obtained in Asia. Here, insufficient measurements preclude confident estimates of the total snow volume. A clear example is the Himalaya, where the local variability in snow conditions is huge and the few (6) clustered measurement sites in Langtang Valley, Nepal, do not represent even the cross-masked Sentinel-1 snow volume retrieval, much less the area-wide mountain range Sentinel-1 retrieval. This highlights the need for extending measurement networks in High Mountain Asia, where water resources from snowmelt are critical[8,49]. At the same time, it also stresses the value of the frequent and systematic C-band SAR observations from Sentinel-1 to quantify snow depth in areas where expanding the network is difficult owing to extreme altitude, poor accessibility or safety.

## Discussion

The Sentinel-1 snow depth retrievals are primarily depending on the cross-polarized backscatter measurements. Temporal variations in $\sigma_{vh}^{0}$ with snow depth evolution have, to our knowledge, not been investigated before with C-band satellite observations. Only few studies deployed tower-mounted radar instruments, with contradictory results. Overall, our study aligns well with the scatterometer measurements over a site in Michigan, USA, revealing an increase in $\sigma_{vh}^{0}$ with an increase in snow depth[26].

However, the scatterometer measurements showed signs of saturation for depths exceeding ~60 cm, which is not observed in the Sentinel-1 measurements. Potential causes for this discrepancy include the use of artificial snow in the scatterometer experiments (characterized by a homogeneous layer of snow composed of small, rounded particles and the absence of snow melt-freeze metamorphism), or differences in the spatial support of the measurements (18 m × 30 m for the scatterometer versus 1 km² for the processed Sentinel-1 data). In strong contrast, a (minor) decrease in $\sigma_{vh}^{0}$ with increasing snow depth was observed from scatterometer measurements over a site in the Swiss Alps[27]. An inverse relationship can occur with site-specific ground, vegetation and snow conditions, if the attenuation of ground scattering by the snowpack is stronger than the scattering contribution from the snowpack[25]. However, the results of the study in the Swiss Alps could also be impacted by the backscatter measurement principle: the total backscatter was calculated by integrating the scattering contributions from a few dominant surfaces, i.e., the snow surface, the ground surface and/or horizontal layers within the snowpack, and was thus not including the multiple scattering[27].

In agreement with the scatterometer measurements in Michigan[26] and with our Sentinel-1 observations, radiative transfer model simulations generally indicate an increase in $\sigma_{vh}^{0}$ with snow depth. Recent model developments tend towards a relatively weak dependence on frequency[34]. A critical aspect in this context is the development of state-of-the-art, theoretically-based radiative transfer models, which allow for simulating volume scattering from snow, represented by clustered, non-spherical particles[17,33–35]. This presents a major improvement over conventional solutions for scattering from individual, spherical snow

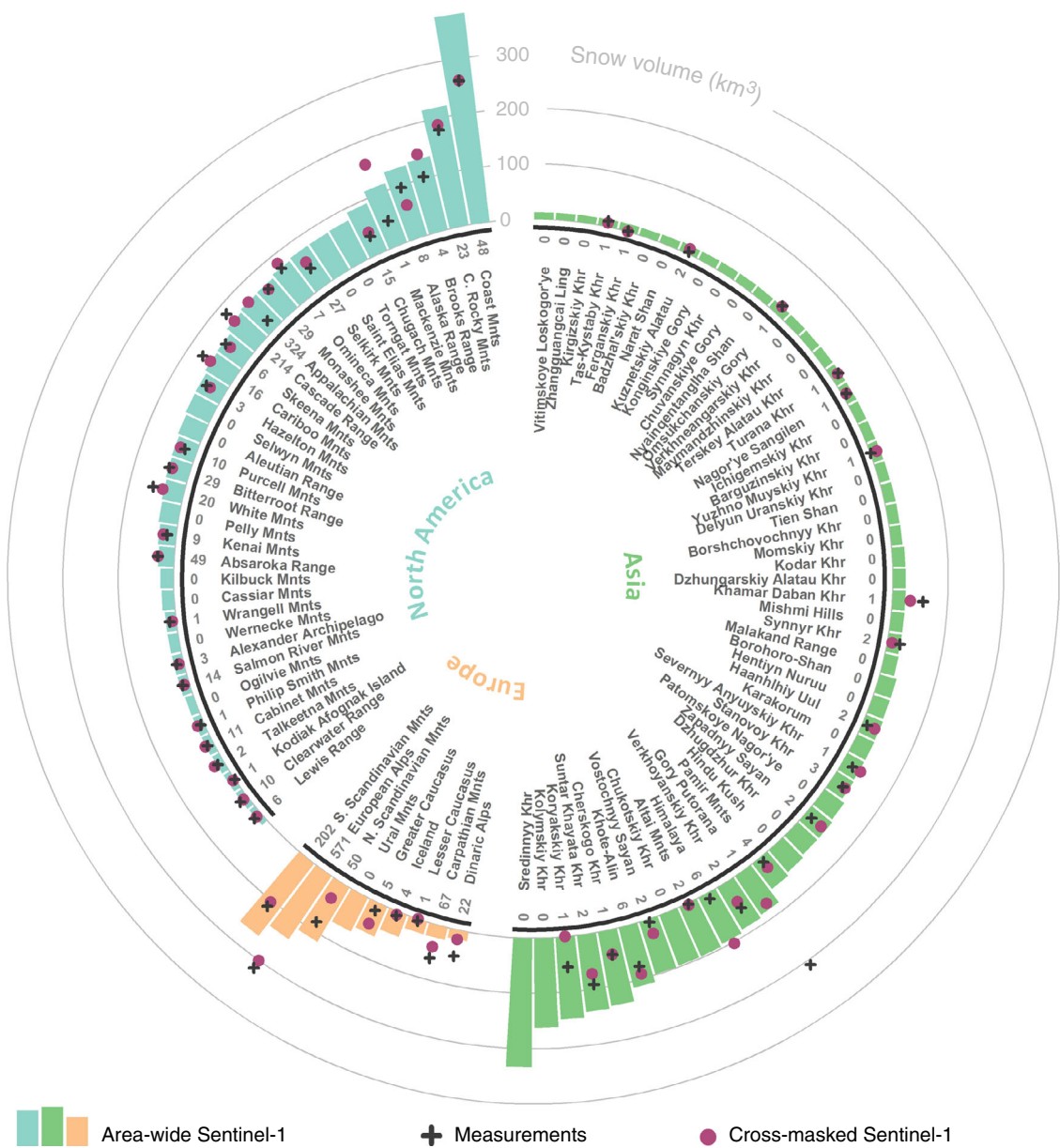

**Fig. 8** Snow volumes for the top 100 snowiest mountain ranges. The snow volumes (km³) are averaged estimates over February 2018 and exclude glaciated areas. Bars represent the area-wide Sentinel-1 estimates averaged over the entire mountain range, black crosses the average of in situ measurements, and purple dots the corresponding average of cross-masked Sentinel-1 estimates, averaged only over grid cells that include in situ measurement sites. The number of in situ measurement sites, if any, is indicated next to the mountain range name. Source data are provided as a Source Data file

particles using Rayleigh theory, assuming that scattering is proportional to the fourth power of frequency and third power of grain size[35]. Most of these advances in radiative transfer modeling were thus far focusing on microwave measurements at higher frequencies (e.g., X-band and Ku-band). We encourage future modeling efforts to unravel the different mechanisms that cause the demonstrated sensitivity to deep snow at C-band.

To further improve the Sentinel-1 retrieval algorithm, we recommend future tower-mounted radar experiments, for instance to investigate the impacts of snow wetness on $\sigma^0_{vh}/\sigma^0_{vv}$. Such measurements would support a quantitative assessment of the uncertainty in the snow depth retrievals caused by wet snow, which in turn may result in a correction for snow liquid water content in the retrieval algorithm. In fact, previous studies demonstrated the retrieval of the snow wet-dry state[23] (and even liquid water content[50]) directly from C-band SAR observations.

Wet snow maps derived from Sentinel-1 could be provided as auxiliary information (quality flag) with the snow depth retrievals. Similarly, the retrievals may be further refined by accounting for the temporal evolution in snow microstructure[10]. For instance, a climatological snow classification with associated snow microstructure and density estimates is available at the global scale[46,47] and could be used to improve the parameterization. Further research is also recommended to investigate the retrieval performance in glaciated areas. Currently, data from the Randolph Glacier Inventory 6.0[48] are processed and provided with the snow depth retrievals to allow for an optional masking of glaciated areas.

In the future, sensor synergies can be exploited by merging our data over mountain areas with passive microwave observations[10–12] outside these areas, providing global coverage, or with active observations at higher frequency (e.g., X-band or Ku-band) that are

likely more favorable for areas with shallow snow[17–19]. The combination with estimates from SAR interferometry[20,21] could further improve the representation of temporal changes. Finally, the assimilation of the snow depth retrievals into a land surface model could yield improved and continuous (in time and space) global estimates of snow depth and mass that may also improve the initialization of numerical weather and climate models and thus improve predictions[51]. For such applications, the long-term continuity in C-band backscatter measurements with the Sentinel-1 constellation and the RADARSAT Constellation Mission is a strong asset, offering the systematic observations that are required to improve the representation of the cryosphere.

## Methods

**Study domain.** We processed Sentinel-1 observations for the period September 1, 2016 through August 31, 2018 over the Northern Hemisphere mountain areas north of 20°N (Fig. 1a). To delineate mountainous areas, we used the Global Mountain Biodiversity Assessment (GMBA) inventory[30], providing features such as geographic coordinates, area, name, and bioclimatic region for more than 1000 individual mountain ranges in the world, which are classified based on the combination of elevation and ruggedness.

**In situ measurements.** Daily, point-scale measurements of snow depth were assembled over mountainous areas for the period September 1, 2016 through August 31, 2018. The measurements originate from a variety of sources, including regional, national and global networks (Table 1). Quality control was applied to the measurements by eliminating values higher than twice the 90th-percentile of the time series (after excluding zero values), and eliminating sites with less than 3 reported values during the evaluation period. The measurement locations were projected onto the 1 km² global cylindrical Equal Scalable Earth version 2.0 (EASE-2) grid[52]. Where several locations fell within the same grid cell, their time series measurements were averaged. This resulted in 4175 unique grid cells with in situ measurements (Fig. 1b).

**Sentinel-1 data processing.** The processing of the ground-range detected (GRD) Sentinel-1A and Sentinel-1B Interferometric Wide Swath (IW) $\sigma^0$ measurements in vv-polarization and vh-polarization was performed using Google Earth Engine's Python api. We applied the standard Sentinel-1 processing techniques, including thermal noise correction, radiometric calibration, and range-Doppler terrain correction (Fig. 2). Further, the $\sigma^0$ data with native 5 m × 20 m resolution and 10 m × 10 m grid spacing were resampled (by averaging in linear scale) and projected onto the 1 km² global cylindrical EASE-2 grid. The 1 km² scale of the resampled Sentinel-1 data matches the resolution of the auxiliary information on snow and land cover used in the retrieval algorithm (Section Sentinel-1 snow depth retrieval algorithm). Moreover, the averaging reduces the speckle noise inherent in radar observations and improves the signal-to-noise ratio, which is typically low for the cross-polarized $\sigma^0_{vh}$ data. Future research will investigate the potential of retrieving snow depths at a higher spatial resolution.

Each Sentinel-1A and Sentinel-1B satellite has an exact 12-day repeat cycle, with 175 orbits per cycle. In a given 1 km² EASE-2 grid cell, the Sentinel-1 observations from the different orbits within one repeat cycle have different incidence angles, ranging from 29.1° to 46.0° relative to a flat surface. Over mountain areas, the impact of the incidence angle on $\sigma^0$ may be large, e.g., when a terrain slope is facing towards or away from the sensor line of sight in ascending and descending overpasses. To account for the effect of the incidence angle, we separated the $\sigma^0$ values pertaining to each of the 175 orbits in a 12-day cycle (repeated for every cycle, with identical incidence angles for each of the orbits). The static bias between $\sigma^0$ values from different orbits was then removed by rescaling the mean $\sigma^0$ of each orbit to the overall mean (i.e., of the entire $\sigma^0$ time series, including all orbits) and applying this mean correction to the individual $\sigma^0$ measurements. The accuracy of the incidence-angle normalization increases as more observations become available. Outliers were removed by excluding values that are 3 dB above the 90th-percentile or 3 dB below the 10th-percentile of the time series. Four sub-sets of Sentinel-1 data (i.e., ascending (6 p.m.) and descending (6 a.m.) data from Sentinel-1A and Sentinel-1B) were pre-processed separately and combined into a single Sentinel-1 dataset.

Both Sentinel-1 satellites share the same orbital plane with a 6-day offset, thus the two-satellite constellation offers an exact 6-day repeat cycle. However, the observation frequency during the period considered here varies from as much as ~daily (for certain areas in Europe) to every ~2 weeks (Fig. 1a), depending on the latitude, the availability of ascending and descending orbits, the availability of Sentinel-1B data (reaching full capability in early 2017), and the evolving Sentinel-1 observation acquisition strategy. Prior to 2017, observations can (regionally) be irregular in time, with extended no-data gaps. For instance, relatively few observations were available over parts of the USA and Canada for September-December 2016 (Fig. 3a–c). We did not process any Sentinel-1 data taken before September 2016 owing to the extremely limited coverage outside Europe.

**Sentinel-1 snow depth retrieval algorithm.** The snow depth (m) retrieval algorithm relies on an empirical change detection method applied to the Sentinel-1 measurements of the cross-polarization ratio ($\sigma^0_{vh}/\sigma^0_{vv}$; in dB). As auxiliary input data, it uses 1 km² snow cover (SC; 1 if present or 0 if absent) from IMS and fractional forest cover (of evergreen species) from the 1 km² global consensus land cover dataset[53]. Firstly, a change detection index (hereafter referred to as snow index, SI; in dB) is calculated for each location $i$ and time step $t$. This index links the temporal changes in Sentinel-1 $\sigma^0_{vh}/\sigma^0_{vv}$ with the accumulation or ablation of snow:

$$\text{SI}(i,t) = \begin{cases} \max\left(0, \left[\text{SI}(i,t-1) + \sigma^0_{vh}/\sigma^0_{vv}(i,t) - \sigma^0_{vh}/\sigma^0_{vv}(i,t-1)\right]\right) & \text{if } \text{SC}(i,t) = 1 \\ 0 & \text{if } \text{SC}(i,t) = 0 \end{cases}$$

(2)

where the maximum operator avoids negative indices. Secondly, the SI is rescaled into snow depth (SD; in m) as:

$$\text{SD}(i,t) = \left(\frac{a}{1 - b\text{FC}(i)}\right)\text{SI}(i,t)$$

(3)

where the parameters $a$ (in m dB⁻¹) and $b$ (dimensionless) are constant in space and time, and $\text{FC}(i)$ is the evergreen forest cover fraction (dimensionless). Forests typically attenuate snow backscatter[54]: they scatter part of the incoming energy, thus preventing it from reaching the snow, and part of the signal backscattered from the snow, thus preventing it from reaching the satellite sensor. This attenuation is corrected as established coarse-scale passive microwave retrievals[12], with the same parameter value $b = 0.6$. The performance of the algorithm can potentially be further improved by optimization of $b$. Moreover, vegetation dynamics can have a very similar impact as snow on backscatter[55]. Therefore, the snow depth retrievals are masked when climatological leaf area index (LAI) from the Moderate Resolution Imaging Spectroradiometer (MOD15A2) exceeds 25% of the dynamic range in the LAI climatology time series. Finally, the snow depth retrievals based on Eq. (3) are smoothed to further reduce the impact of Sentinel-1 observation noise and to reduce short-term, high-magnitude fluctuations[10]. This is done using linear inverse distance weighting with a 2 km radius in space and a 10 day radius in time. Depending on user requirements, the smoothing can be adjusted or switched off (e.g., for data assimilation).

The parameter $a$ (Eq. (3)), impacting the magnitude of the Sentinel-1 snow depth retrievals, was estimated based on in situ measurements. More specifically, $a$ was optimized by minimizing the MAE between the times series of the global average snow depth measurements and corresponding Sentinel-1 retrievals. For the optimization, we excluded the time periods from March to August to limit the impact of wet snow, which is not yet accounted for in the algorithm and would likely increase errors in the retrievals. The in situ measurement locations (Section In situ measurements) were randomly sampled in two subsets with equal size: one for calibration and the other for validation. The calibrated value of $a$ is 1.1 m dB⁻¹.

**Validation.** Several performance metrics are calculated to validate the Sentinel-1 snow depth retrievals using in situ measurements. As the retrieval algorithm includes the rescaling parameter $a$ (Eq. 3) that is optimized based on a subset of the in situ measurements (i.e., the calibration subset), some metrics are only calculated using the remaining subset of the measurements (i.e., the validation subset) to provide an independent assessment. The spatial Pearson correlation coefficient ($R_s$; dimensionless) is calculated between monthly-averaged point-scale in situ snow depth measurements and Sentinel-1 retrievals for corresponding grid-cell locations ($N = 4175$). The calculation of $R_s$ uses data from both the calibration and validation subsets, because the calibrated parameter $a$ does not impact the metric. Similarly, temporal correlation coefficients ($R_t$; dimensionless) between time series of in situ snow depth measurements and corresponding Sentinel-1 retrievals are also calculated for sites pertaining to both the calibration and validation subsets for the same reason. However, we included only sites ($N = 1800$) with more than 25 (i.e., once every 4 weeks, on average) non-zero values, since the calculation of $R_t$ is sensitive to the sample size. The MAE and bias (mean of retrievals minus measurements) are calculated only for sites ($N = 2000$) of the validation subset with at least one non-zero value, as these metrics are less dependent on the sample size. This also allows evaluating the performance for sites with occasional snow or few reported values (such as the monthly USDA Aerial Markers).

We use point-scale in situ measurements for evaluating the performance of the Sentinel-1 snow depth retrievals. However, this likely provides a conservative estimate of the true retrieval performance. Especially in mountain areas, the point-scale snow measurements do not necessarily resemble the 1 km² grid-cell average snow conditions represented by Sentinel-1. The local variability in conditions can be large due to differences in elevation, slope and aspect, as well as wind and vegetation impacts on snow distribution[56]. In situ sites are preferentially located in relatively flat and non-forested terrain that is often not representative of the large variations in slope and forest cover in the surrounding area[36]. Consequently, in situ measurements often poorly represent the snow accumulation and melt rates on nearby slopes[9,57]. Ideally, the validation of the Sentinel-1 retrievals would therefore be performed using data that can be processed at the matching scale, such as gridded estimates from airborne lidar[5,14], regional model simulations[41] or high-resolution model reanalysis[58]. However, such information is not available at the

global scale, which limits this study to the comparison with in situ measurements, despite the representativeness differences.

**Reporting summary**. Further information on research design is available in the Nature Research Reporting Summary linked to this article.

## Data availability

The Sentinel-1 snow depth retrievals are available online at https://ees.kuleuven.be/project/c-snow. The source data for Figs. 1a, 4, 5, 7a, d correspond to the Sentinel-1 snow depth retrievals provided through the above-mentioned website. The source data underlying Figs. 7c, f and 8 are provided as a Source Data file.

## Code availability

The source code is available from the corresponding author upon reasonable request.

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

## Acknowledgements

This work was funded through the BELSPO SNOPOST and C-SNOW projects. Part of the study was performed by H.L. at NASA/GSFC, with computational resources provided by the NASA High-End Computing Program through the Center for Climate Simulation. R.R. and M.G. were supported by the NASA Terrestrial Hydrology program. Sentinel-1A/B data are from the ESA and Copernicus Sentinel Satellites project and were processed using Google Earth Engine.

## Author contributions

H.L. and G.D.L. designed the study. M.De. processed the Sentinel-1 data. H.L., H.M., R.R., L.B., I.B., M.G., E.K. and G.D.L. contributed to the Sentinel-1 snow depth retrievals. Pd.R., M.Du., W.I., T.J., I.K., C.M., T.S., J.S. provided the in situ snow depth measurements. All co-authors contributed to the writing of the paper and the discussion and interpretation of the methods and results.

## Competing interests

The authors declare no competing interests.
