## [Peer Review File · Nature Communications]

Reviewers' comments:

Reviewer #1 (Remarks to the Author):

Review of paper

Lievens et al. "Snow depth variability in the Northern Hemisphere mountains observed from space"

by: Juha Lemmetyinen

General comments

The study by Lievens et al. presents a method for retrieving snow depth from C-band, dual-polarization Sentinel-1 radar backscatter data. The basis of the retrieval is on the assumption that C-band cross-polarized backscatter is sensitive enough to snow volumetric properties to provide a basis for the retrieval, with an increase in cross-polarized backscatter correlating with increases in the amount of snow in the radar signal propagation path. Previously, C-band has been largely disregarded in retrievals of dry snow properties (depth, SWE) due to the acknowledged low volume scatter presented by dry snow particles at this relatively long wavelength. The authors note that while their study corroborates previous findings regarding co-polarized backscatter, their analysis shows that such a sensitivity can be found for cross-polarization, and that the sensitivity can be further distinguished by analyzing the ratio of cross- and co-pol backscatter. While early experimental studies (notably by Kendra and others) provided some indications of usability of C-band cross pol for this purpose, the authors note that S1 is the first satellite system providing these data readily on a regular basis and over large areas (access to data from previous polarimetric systems such as RadarSAT-1, -2 has been more restricted), which may be the underlying reason why these relations have not been found previously.

The study should be of very high interest to the snow remote sensing community, as developing a method or a sensor suitable for large scale retrieval of snow depth over mountains has been long sought after. As stated by the authors, airborne lidar may offer the best solution over individual watersheds, but having mapping capability from an existing satellite system such as Sentinel-1 would provide unprecedented coverage. Even though the accuracy achieved by the presented method may not address all user requirements in terms of spatiotemporal resolution and accuracy (and it is a bit unclear from the manuscript how the achieved accuracy relates to these requirements), it would nevertheless present a significant step further in achieving this goal. Successfully employing C-band radar backscatter intensity for snow depth mapping certainly goes against present thinking in the field, but the authors have built a convincing case to support their claims. That said, the method is presented in sufficient detail for other groups to be able to verify the results at least on a basic conceptual level.

Main technical concerns I have are related to the potential inclusion of glacier areas in Figure 3 (I think glaciers should be masked by IMS but not sure) and the forest fraction correction, which is adapted "as is" from passive microwave. I think this should be adjusted, or at least justified better.

The study is restricted to snow over mountains in the Northern Hemisphere; while this leaves out most of the snow covered area, it gives focus and a clear motivation for the study by concentrating on crucial areas not covered by other satellite remote sensing methods such as passive microwave. A limitation is that only two years of data are applied, since the inter-annual variability of snow properties affecting radar and radiometer signatures is known to be high; on the other hand, covering all the major mountain ranges in the Northern Hemisphere provides confidence that different snow conditions are covered.

In general, the manuscript is clear, well written and the main part is sufficiently concise while not leaving out any crucial analysis. I have some comments mainly on the supplementary section, where I would wish the authors better justify, or adjust, some of the approaches they have taken. If these concerns can be met I would recommend the study for publication.

Specific comments

1. P5 lines 110-114 and section Sentinel-1 SD evaluation in the Supplement: Assuming the achieved accuracy metrics presented here can be generalized to represent the accuracy of the retrieval approach, how do these metrics relate to requirements for snow depth retrieval accuracy over mountains (e.g. from WMO OSCAR, or other sources)? This should somehow be commented. What about SWE, assuming there are further errors from the necessary estimation of snow density? How does the accuracy performance stand in comparison to other methods (e.g. as defined by Dozier et al., 2016)?

2. P6 mountain range snow volumes: I could not find a mention whether glaciers were masked or not from the retrieval, and thus the total volume estimates presented here; IMS probably masks out glaciers, so the glacier mask comes out essentially from eq.2? If so this could be explicitly mentioned to avoid confusion.

3. P8 lines 192-194: Other C-band systems (e.g. RCM) could be acknowledged as providing potentially complementary data to S1 in the future.

4. Lines 350-354. The justification for 1 km² scale: while this represents an order of magnitude of improvement vs. present satellite products, it would be necessary to comment if other spatial resolutions were experimented, and if so, were results better or worse? Referring to auxiliary data seems a bit artificial as a reason since e.g. other land cover data is available at a resolution exceeding this. Did you analyze the optimal number of looks from the radiometric point of view? The scaling issue is also relevant because of the point-wise data you have applied for training and validation, as well as small scale changes in snow structure. If 1km² is the optimal (or close to optimal) spatial resolution, could it be that in addition to radar speckle, this is also where small-scale structural changes in snow affecting the radar signature are optimally "averaged out", while the individual point measurements of SD are still representative? All this said, I feel it would be necessary to comment and justify the choice of spatial scale further.

5. P19, line 418: It is difficult to be convinced of the claim that the cross- co-pol ratio "shows a clear correlation with snow depth", since snow depth is not presented in figure S2. The sentence should be modified, or better, e.g. the temporal change of average snow depth over the corresponding area added to the FigS2.

6. P19, line 419. You state that applying the ratio "may partially mitigate impacts from temporal changes". I assume you tried also correlating only the cross pol? And apparently, the cross-co pol ratio yielded better results? This should be explicitly stated, now the justification for using ratio in place of just the cross pol backscatter is not very robust.

7. P20, lines 426-439. It is not entirely clear to me whether you applied both am and pm orbits in your analysis. I am assuming both, but would it not be more prudent to apply only am orbits, in particular in the light of your discussion here? You would lose some temporal resolution, but have less chance of wet snow? Or, did you consider applying a wet snow mask, based on mapping wet snow with S1 itself, following e.g. temporal change detection methods presented by Nagler et al. (reference 23 in the manuscript)? In the least, these methods could be commented as they could be highly complementary to your snow depth retrievals. In this context I am also not sure of what is meant by the last sentence in the paragraph (lines 438-439): what do you mean by wet snow observations being "evaluated throughout the paper"? Do you mean to say that in practice, you included all data, with no regard to possible wet snow?

8. P21 lines 454-459. Fine for using the basic index modifier in eq (3) related to vegetation; applying exactly the same parameter b as for passive microwave, however, would require some further justification. This is because the observation geometries and wavelengths between S1 and passive microwave sensors are very much different. Did you experiment adjusting parameter b? While optimizing both a and b simultaneously might not yield satisfactory results, taking forested grid cells with a range in FC, but more or less equal SD (+/- 5 cm, for example) from your training

dataset, and attempting to optimize b might provide another optimal value?

9. P21 line 469-P22 line 478. As the authors point out, experimental data making use of cross-polarized C-band data should be collected to study the effect of wet snow; also it would be necessary to further corroborate the claims presented in the paper, and notably to increase understanding on the cross-polarized response at different wavelengths. This could be made more clear. In this regard, authors could perhaps state more explicitly that such data has been collected in the past, notably by Kendra and others as well as Strozzi and others – both references can be found in the paper. In particular, the authors should highlight (briefly) the main findings of those studies, and why they do not fulfill the requirements needed to corroborate the findings here. The study using artificially accumulated snow by Kendra and others did, after all, find indications of increasing cross-polarized backscatter at C-band, while measurements by Strozzi and others did not (possibly due to a high level of ground backscatter). Both experiments were rather short in time and/or limited to a few sites, and thus limited in representing natural snow in different stages of metamorphism.

Reviewer #2 (Remarks to the Author):

Review of "Snow depth variability in the Northern Hemisphere mountains observed from space", by Lievens et al.

The authors find that C-band cross-pol SAR contains information about snow accumulation in mountains. They present an algorithm and calibrate global parameter values between in situ data and SAR. The validation demonstrates that the retrievals have adequate skill to be useful in mountain areas.

This timely, important finding should be published; it is of broad interest, with the mere fact of it working appealing to remote sensing of snow community, and the broader perspective of measuring snow in global mountains appealing to a wider audience. The manuscript is generally well-written, and the analysis is sound. All of the edits I'm asking for relate to better explaining the measurement principle in the context of existing theory and observations which have not anticipated the result shown here. Honestly, I think the paper would be better if the authors said that it is not yet clear why C-band cross-pol is sensitive to such deep snow, but that this paper provides empirical evidence that this method works.

Specific Comments

1. Two in situ observation studies are referenced in explaining the measurement principle references 19, and 26, King et al 2005 and Kendra et al 1998, respectively. Kendra et al. find that C-band cross-pol backscatter (see their Fig 7a) seems to saturate at about 50 cm, which is inadequate in the mountains. How do the authors fit their result with the much lower saturation depth observed by Kendra et al 1998? A third paper deserves mention: Chang et al. 2014. They look at X- and Ku- data from NoSREx, an important in situ radar snow experiment in Finland. They document cross-pol data with some response to snow depth at Ku-band, but almost no response at X-band. Why wouldn't the expectation be that there is even less response at C-band?

2. Most theory does not adequately treat cross-pol: e.g. Shi & Dozier (2000) (ref 25 in the submission) state that their approach for analyzing Sir-C only handles co-pol as it only treats first order back-scattering. I think it would be nice to make this point in the manuscript. Models capable of simulating cross-pol should also be mentioned. Xu et al. (2012) use a more sophisticated model and show a decreasing cross-pol response to snow depth as frequency decreases from 17 to 10 GHz; while they do not show C-band results, the cross-pol response in that case would be nearly zero by extension. Du et al. 2010 and Yueh et al. 2009 hypothesized that (higher-frequency) cross-pol response was likely due to grain shape asymmetry. I think that

- pointing to these hypotheses as to why the cross-pol response is observed should be added.
3. Line 28-29: I think this overstates the finding. Sentinel-1 C-band cross-pol is clearly correlated with snow depth. But I think the abstract, while it has to be brief, ought to hint that this is a purely empirical approach and that the physics behind why it works are not well understood.
 4. Line 66-67: I think you should mention here that the method relies on calibrating the radar response.
 5. Line 79: ref 19 by King does not show cross-pol response > co-pol response at Ku band, in my understanding.
 6. Line 88: I think again you need to say that you are calibrating to in situ data, and that the measurement principle is not yet well understood.
 7. Line 92-102: In this section, I personally would think that showing a few select sites with in situ data in the supplement or even the main paper would be far more convincing than what is shown, which are the radar data averaged over large regions in Fig S2, and R values in Fig S4. This to me would go a long ways to showing that the signal is responsive to deep snow.
 8. Line 179-180: It is not clear why the observations are sensitive to deep snow. I would reword.

References

Chang et al., 2014. Dense Media Radiative Transfer Applied to SnowScat and SnowSAR, IEEE JOURNAL OF SELECTED TOPICS IN APPLIED EARTH OBSERVATIONS AND REMOTE SENSING, VOL. 7, NO. 9, SEPTEMBER 2014.

Du et al. 2010. Comparison between a multi-scattering and multi-layer snow scattering model and its parameterized snow backscattering model, Remote Sensing of Environment 114 (2010) 1089 – 1098.

Xu et al. 2012. Electromagnetic Models of Co/Cross Polarization of Bicontinuous/DMRT in Radar Remote Sensing of Terrestrial Snow at X- and Ku-band for CoReH₂O and SCLP Applications. IEEE JOURNAL OF SELECTED TOPICS IN APPLIED EARTH OBSERVATIONS AND REMOTE SENSING, VOL. 5, NO. 3, JUNE 2012

Yueh et al. 2009 Airborne Ku-Band Polarimetric Radar Remote Sensing of Terrestrial Snow Cover. IEEE TRANSACTIONS ON GEOSCIENCE AND REMOTE SENSING, VOL. 47, NO. 10, OCTOBER 2009

Reviewer #3 (Remarks to the Author):

My comments are in the attached pdf file. It is an original and very interesting manuscript that will interest a large audience but it brings more questions than answers. The authors need to convince me that the proposed algorithm is deriving the snow depth in mountainous areas. I look forward to read their reactions to my comments. Sincerely yours.

Reviewer #3, continued (Remarks to the Author):

References:

Studies has been done in some mountainous areas, Alpes, Rockies Canadian and Libanon. However, the snow wetness was a problem.

CORBANE*, C., SOMMA, J., BERNIER, M., FORTIN, J.P., GAUTHIER, Y*., DEDIEU, J.P. (2005). Estimation de l'équivalent en eau du couvert nival en montagne libanaise à partir des images RADARSAT-1. Conférence Snow Hydrology of Mediterranean Regions à Beyrouth, Liban, 15-17 décembre 2002, Journal des Sciences Hydrologiques, 50(2) : 355-370.

J.-P. Dedieu, N. Besic, G. Vasile, J. Mathieu, Y. Durand and F. Gottardi, Dry snow analysis in alpine regions using RADARSAT-2 full polarimetry data. Comparison with in situ measurements. IEEE International Geoscience and Remote Sensing Symposium (IGARSS'14), July 13–18, Quebec, Canada, pp. 3658–3661, 2014.

Arnab Muhuri, Surendar Manickam, Avik Bhattacharya, Snehamani, "Snow Cover Mapping Using Polarization Fraction Variation With Temporal RADARSAT-2 C-Band Full-Polarimetric SAR Data Over the Indian Himalayas", Selected Topics in Applied Earth Observations and Remote Sensing IEEE Journal of, vol. 11, no. 7, pp. 2192-2209, 2018.

Also, pertinent references are missing:

ALGOSNOW - Contract No. 4000103180/11/NL/CT: Algorithms for Snow and Land Ice Retrieval using SAR data. ESA Report, 2013.

Rott, H. et al. "Development of snow retrieval algorithms for CoReH2O – Final Report, ESA-ESTEC Contract 22830/09/NL/JC, 2011.

Macelloni G, Brogioni M, Montomoli F, Fontanelli G. 2012. Effect of forests on the retrieval of snow parameters from backscatter measurements. Eur J Remote Sensing, 45: 121–132, 2012.

Figure 1: There are glaciers in the Mountainous areas. Those areas should be localized. It seems that the areas identified as "Mountainous, no snow" in Figure 1 included the glaciers.

Figure 3: I don't understand the message in this Figure. What is the difference between Cross-marked Sentinel and Areas Wide Sentinel?

Sentinel-1 data processing:

Line 375-377: I did not understand the final processing done after the incidence angle normalization? What were the range of those incidence angles?

A schema would help to understand all the steps of the image processing. Those done by ESA-Copernicus (format of the images downloaded), those done by the authors.

Algorithm:

Line 450: Equation 3 shown that you take care of the evergreen forest cover fraction (in %) in a given pixel of 1 km, using an attenuation constant (b). However, elsewhere in the manuscript, it is indicated that you applied the algorithm above the tree line (2500 m).

I would be curious to see how your Snow Indices values (equation 2) are correlated with the Snow depth? The correlation could be better than SD for pixels above the tree line.

Line 456-458: "Finally, the SD observations based on Eq. (3) are smoothed to further reduce the impact of Sentinel-1 observation noise. This is done using linear inverse distance weighting with a 2 km radius in space and a 10 day radius in time".

I don't agree that there are still a noise in the Sentinel-1 data after all the smoothing done in the processing of the data (filtering and resampling to 1km). As shown in Figure S2b, there are no variation in the VV signal during the summer (-8 dB) and one would expect that raining and drying period would affect the backscattering signal from the ground (soil- vegetation) unless the values shown are for rocky area. Also, it would be expect that the soil freezing (at least in the beginning of the winter when the snow cover is shallow) a decreasing in backscattering will be recorded in VV polarization but the signal in winter is also stable at -8 dB.

Line 464: For the optimization of parameter a, you excluded the time periods from March to August to avoid impacts from wet snow. However, it seems that you applied equation 2 and 3 to Sentinel data acquired in the same period (Figure S5)?

Line 470-474: To further improve the retrieval algorithm, your recommend future research to investigate the impacts of snow wetness on $\sigma^0_{vh}/\sigma^0_{vv}$, for instance using tower-mounted radar measurements and including local observations of snow liquid water content and snow density in the in the retrieval algorithm. I don't think that you can hope to get better results in including more parameters that vary in spatially and temporally like snow liquid water content and snow density. Those information will be difficult to get globally as they are already scarce locally. Further, as shown in Figure S2d, in 2017, between March 1st and June 1st, the range of the Ratio is only 2 dB.

Then, I am not comfortable with you recommendation (line 470-474) which is in contradiction with one of your statement (line 420) that taking the ratio may partially mitigate the impacts from temporal changes in the ground surface, vegetation, or snow microstructure conditions. From my experience, the ratio is mainly reducing the slope effects in Mountain Area and also the variation in local incidence angles which are link and then vegetation and snow.

Regarding, the availability of the snow measurements in VH and VV polarisation for different frequencies, some data are available in some recent ESA reports about snow algorithms for SWE and the NASA SnowEX.

Line 494-498: I agree with the authors that ideally, the validation of the Sentinel-1 observations would therefore be performed using data that can be processed at the matching scale, such as gridded estimates from airborne lidar, regional model simulations or high-resolution model

reanalysis but that such information is not available at the global scale. Then, the authors should be more cautious in their conclusions regarding the correlation between the Sentinel-1 prediction model (equation 2 and 3) and the measured Snow Depth as well in the interpretation of Figure S5.

In particular for the snow melt period (March 1st- June 1st) , the decrease in backscattering is linked on the wet snow absorption of the signal in both VH and VV polarisation and that absorption increases with the warming of the air temperature and increases on the liquid water content at the air/snow interface. This decreasing of the snow signal correlated indirectly with the melting of the snow cover and the decreasing of the snow depth. Then, I don't agree with your interpretation (line 420-424) that the ratio decreases because of the relatively higher decrease rate of σ_0^{vh} , caused by the reduced volume scattering (and depolarization) in an increasingly shallow snowpack.

Besides, the backscattering is increasing in early summer (approximately in June) in both polarisations with the decrease of the snow cover fraction (in %) within a 1km pixel and the increase contribution of the wet soil and the vegetation. Then, in the late snow season the VH/HH ratio don't vary much (Figure S2D). I am surprised that you derived SD values that correlate with the field data for that period (June-July), although it is difficult to see on the graphs of Figure S5 and the correlation shown are globally for the two years and not by season.

I am more comfortable with the link between the increase of the Ratio in winter with the increase in volume scattering from the snowpack in VH polarization where there is no coniferous tree and the snowpack is dry as this behavior has been reported elsewhere for C-band and X-band. However, other environmental parameters (unfrozen ground) or images processing could explain the good correlation. May be the huge dataset used for the optimisation (a parameter) but limited to two winters could explain the good correlation and the relatively small RMS with field measurements which are also not always representative of the heterogeneity of the snowpack.

In conclusion, the authors still need to convince me that their algorithm is estimating the snow depth in mountainous areas.

Reviewer #1 comments (Dr. Juha Lemmetyinen)

General comments

The study by Lievens et al. presents a method for retrieving snow depth from C-band, dual-polarization Sentinel-1 radar backscatter data. The basis of the retrieval is on the assumption that C-band cross-polarized backscatter is sensitive enough to snow volumetric properties to provide a basis for the retrieval, with an increase in cross-polarized backscatter correlating with increases in the amount of snow in the radar signal propagation path. Previously, C-band has been largely disregarded in retrievals of dry snow properties (depth, SWE) due to the acknowledged low volume scatter presented by dry snow particles at this relatively long wavelength. The authors note that while their study corroborates previous findings regarding co-polarized backscatter, their analysis shows that such a sensitivity can be found for cross-polarization, and that the sensitivity can be further distinguished by analyzing the ratio of cross- and co-pol backscatter. While early experimental studies (notably by Kendra and others) provided some indications of usability of C-band cross pol for this purpose, the authors note that S1 is the first satellite system providing these data readily on a regular basis and over large areas (access to data from previous polarimetric systems such as RadarSAT-1, -2 has been more restricted), which may be the underlying reason why these relations have not been found previously.

The study should be of very high interest to the snow remote sensing community, as developing a method or a sensor suitable for large scale retrieval of snow depth over mountains has been long sought after. As stated by the authors, airborne lidar may offer the best solution over individual watersheds, but having mapping capability from an existing satellite system such as Sentinel-1 would provide unprecedented coverage. Even though the accuracy achieved by the presented method may not address all user requirements in terms of spatiotemporal resolution and accuracy (and it is a bit unclear from the manuscript how the achieved accuracy relates to these requirements), it would nevertheless present a significant step further in achieving this goal. Successfully employing C-band radar backscatter intensity for snow depth mapping certainly goes against present thinking in the field, but the authors have built a convincing case to support their claims. That said, the method is presented in sufficient detail for other groups to be able to verify the results at least on a basic conceptual level.

Main technical concerns I have are related to the potential inclusion of glacier areas in Figure 3 (I think glaciers should be masked by IMS but not sure) and the forest fraction correction, which is adapted “as is” from passive microwave. I think this should be adjusted, or at least justified better.

The study is restricted to snow over mountains in the Northern Hemisphere; while this leaves out most of the snow covered area, it gives focus and a clear motivation for the study by concentrating on crucial areas not covered by other satellite remote sensing methods such as passive microwave. A limitation is that only two years of data are applied, since the inter-annual variability of snow properties affecting radar and radiometer signatures is known to be high; on the other hand, covering all the major mountain ranges in the Northern Hemisphere provides confidence that different snow conditions are covered.

In general, the manuscript is clear, well written and the main part is sufficiently concise while not leaving out any crucial analysis. I have some comments mainly on the supplementary section, where I would wish the authors better justify, or adjust, some of the approaches they have taken. If these concerns can be met I would recommend the study for publication.

We would like to thank the reviewer for providing encouraging and constructive feedback to our work. In the revised manuscript, we have now provided improved justification to the approaches (see in particular responses R.1, C.4-8). We included glaciated areas as a flag into the snow depth retrievals, and removed glaciated areas from the snow volume estimates per mountain range (Figure 3). We also added a paragraph comparing the performance of the Sentinel-1 snow depth retrievals with the literature. Please find more detailed point-by-point responses below.

Specific comments

1. P5 lines 110-114 and section Sentinel-1 SD evaluation in the Supplement: Assuming the achieved accuracy metrics presented here can be generalized to represent the accuracy of the retrieval approach, how do these metrics relate to requirements for snow depth retrieval accuracy over mountains (e.g. from WMO OSCAR, or other sources)? This should somehow be commented. What about SWE, assuming there are further errors from the necessary estimation of snow density? How does the accuracy performance stand in comparison to other methods (e.g. as defined by Dozier et al., 2016)?

R.1, C.1: We have added a paragraph to put the performance of the Sentinel-1 retrievals into context, e.g., compared to the Airborne Snow Observatory and AMSR-E. However, it is important to bear in mind that the conditions of each of these performance evaluations are different. For instance, lidar observations are only assessed locally, whereas AMSR-E focuses on areas outside mountain regions with relatively shallow snow depths (<0.8 m; note that shallow snow depths typically decrease the correlation and reduce the MAE). The following text was added on P.26, L.584: **“For reference, the most accurate snow depth estimates today are likely provided by airborne lidar, offering decimeter-scale accuracy in surface height (either snow-on or snow-off) over local areas¹⁴. As an example, lidar snow depth estimates from the Airborne Snow Observatory have an MAE of ~0.08 m (at the 15 m × 15 m scale) with respect to manual in situ measurements over a relatively flat area near Tioga Pass in the Sierra Nevada, California, USA⁵. The AMSR-E snow depth retrievals, available at global-scale, have correlations of ~0.25 and ~0.41 with estimates from the Snow Data Assimilation System (SNODAS) and WMO in situ measurements¹², and an MAE of 0.20 m against WMO in situ measurements¹⁰. However, a direct comparison with the Sentinel-1 retrieval performance is hampered by the limitation of the AMSR-E retrievals to areas without complex topography and with snow depths typically below 0.8 m.”**

In future research, we plan to perform a direct regional comparison with different types of snow depth (or SWE) estimates, such as high-resolution model simulations, snow reanalysis/reconstruction, and airborne lidar observations.

The WMO Observing Systems Capability Analysis and Review Tool (OSCAR) lists accuracy requirements for snow depth (<https://www.wmo-sat.info/oscar/variables/view/206>). However, these apply to “nowcasting and very short range forecasting” applications, which is not the topic of our study on satellite retrievals primarily to support hydrological applications. Nonetheless, the OSCAR requirements for “nowcasting and very short range forecasting” of snow depth are: an uncertainty threshold of 0.02 m and a spatial resolution threshold of 15 km. We believe that a comparison of the Sentinel-1 performance with the WMO OSCAR

requirement is misleading because one results from the comparison of satellite retrievals with in situ measurements, while the other applies to a model output.

For SWE, OSCAR requirements are formulated for hydrological applications (<https://www.wmo-sat.info/oscar/variables/view/145>) as an uncertainty threshold of 20 mm and a spatial resolution threshold of 10 km. Our 1-km resolution Sentinel-1 observations have a mean absolute error (MAE; against point-scale in situ measurements) of 0.18 m in snow depth. To address constructively the reviewer's comment, assuming for instance a snow density of 300 kg/m³ would lead to an MAE of ~54 mm in SWE at the 1-km scale. The error will reduce when aggregating to a coarser scale (e.g., the 10 km threshold for the hydrology application), and will increase when accounting for uncertainties in the snow density estimates. Thereby, the impact of uncertainties in snow density is likely limited, since snow depth explains most of the SWE variability (Sturm et al., 2010). However, in addition to the fact that our study focuses on snow depth (and not SWE), there are two additional reasons why we refrain from presenting a comparison with the OSCAR SWE requirements. First, the OSCAR requirements are specified for global applications. Most of the global land surface is covered by relatively shallow snow, for which a 20 mm uncertainty requirement is probably realistic. However, this requirement may not be applicable to mountain regions, receiving meters of snow in some places. Second, we believe that estimates of the true Sentinel-1 uncertainty (to be benchmarked against the requirements) should not be derived from comparing grid-scale observations with single point-scale in situ measurements, but rather from a comparison at the matching scale.

Sturm, M. et al. Estimating snow water equivalent using snow depth data and climate classes. *J. Hydrometeorol.* **11**, 1380-1394 (2010).

2. P6 mountain range snow volumes: I could not find a mention whether glaciers were masked or not from the retrieval, and thus the total volume estimates presented here; IMS probably masks out glaciers, so the glacier mask comes out essentially from eq.2? If so this could be explicitly mentioned to avoid confusion.

R.1, C.2: We thank the reviewer for raising this concern. The snow depth retrievals in the previous version of the manuscript included glaciated areas. The IMS data only distinguishes between land with and without snow cover, and sea ice. It does not delineate glaciated areas over land.

Motivated by the reviewer's comment, we have now investigated the impact of glaciers on our retrievals. Therefore, we processed the data from the global Randolph Glacier Inventory (RGI 6.0; <https://www.glims.org/RGI>), by mapping the absence or presence of glaciers (0 or 1, respectively) onto the 1-km² EASE-2 grid used for the Sentinel-1 retrievals. We found that only 8 of the 4175 EASE-2 grid cells with in situ measurements were (at least partially) covered by glaciers. The impact of glaciers on the in situ evaluation of the Sentinel-1 snow depth retrievals was negligible, with an impact on correlation metrics below 0.001.

For some mountain ranges (e.g., the Kenai, Chugach and Saint Elias Mountains, but also the Coast Mountains), glaciated areas were impacting the snow volumes presented in Figure 3. Since we have insufficient in situ measurements in glaciated areas to assess the snow depth retrieval performance, we decided to mask out glaciated areas for the calculation of the associated snow volumes. Figure 3 in the manuscript has been revised accordingly. The following text was added on P.6, L.151: "Figure 3 shows Sentinel-1 observations of the February 2018 snow volumes (in km³) for the top 100 snowiest mountain ranges in the Northern Hemisphere, **excluding glaciated areas**", and to the Figure 3 caption: "**Glaciated areas according to the Randolph Glacier Inventory 6.0⁴¹ are excluded from the estimates**". The added reference 41 is:

RGI Consortium. Randolph Glacier Inventory - A dataset of global glacier outlines: Version 6.0: Technical report, Global Land Ice Measurements from Space, Colorado, USA, <https://doi.org/10.7265/N5-RGI-60> [access date: 03/06/2019] (2017).

The location of the RGI 6.0 glaciated areas will also be added to the distribution of the Sentinel-1 snow depth data as a quality flag. This allows to optionally mask out glaciated areas if required by the application. As an example, Figure R1 illustrates the processed glaciated areas from the RGI 6.0 for the European Alps, on top of the Sentinel-1 snow depth.

Figure R1: **a**, The Sentinel-1 snow depth (m) in the European Alps averaged for February 2018. **b**, The same, with glaciated areas marked in orange.

3. P8 lines 192-194: Other C-band systems (e.g. RCM) could be acknowledged as providing potentially complementary data to S1 in the future.

R.1, C.3: We modified the text on P.8, L.199 to: **“For such applications, the long-term continuity in C-band observations with the Sentinel-1 constellation and the RADARSAT Constellation Mission is a strong asset, offering the systematic observations that are required to improve the representation of the cryosphere”**.

4. Lines 350-354. The justification for 1 km² scale: while this represents an order of magnitude of improvement vs. present satellite products, it would be necessary to comment if other spatial resolutions were experimented, and if so, were results better or worse? Referring to auxiliary data seems a bit artificial as a reason since e.g. other land cover data is available at a resolution exceeding this. Did you analyze the optimal number of looks from the radiometric point of view? The scaling issue is also relevant because of the point-wise data you have applied for training and validation, as well as small scale changes in snow structure. If 1km² is the optimal (or close to optimal) spatial resolution, could it be that in addition to radar speckle, this is also where small-scale structural changes in snow affecting the radar signature are optimally “averaged out”, while the individual point measurements of SD are still representative? All this said, I feel it would be necessary to comment and justify the choice of spatial scale further.

R.1, C.4: The choice of the 1-km resolution was predefined for this work, and we did not test finer resolutions yet. The selection of 1 km was mainly motivated by the spatial resolutions of the auxiliary inputs (snow cover and fractional land cover) used in our processing chain. In support of our reference to auxiliary inputs, it is important to differentiate between land cover and fractional land cover data. The former would indicate

whether or not a certain forest type is present in a grid cell, while the latter provides its fractional cover (in percentage). The use of fractional land cover (forest cover here) allows for a more elaborate correction of the snow depth retrievals. To our knowledge, fractional land cover data is not yet available at finer resolution. In terms of snow cover, MODIS data are available at 500 m resolution, but at the time we designed our processing chain, it was not yet available at that resolution as a cloud-gap-filled daily product (daily data are needed for our application). Our choice for IMS was at that time supported by the team developing the MODIS snow cover product (personal communication with Dorothy Hall; January 2018).

In future research, we will investigate the retrieval at higher spatial resolution. With this, we can explore and compare different sources of (regionally or globally available) land and snow cover datasets, and address the optimal scale. As the reviewer suggests, it will indeed be interesting to investigate this aspect in the context of snow microstructure variability and representativeness differences with in situ measurements.

The following statement is added to the manuscript on P.18, L.377: **“Future research will investigate the potential of retrieving snow depths at a higher spatial resolution”**.

5. P19, line 418: It is difficult to be convinced of the claim that the cross- co-pol ratio “shows a clear correlation with snow depth”, since snow depth is not presented in figure S2. The sentence should be modified, or better, e.g. the temporal change of average snow depth over the corresponding area added to the FigS2.

R.1, C.5: We acknowledge that the lack of corresponding snow depth measurements hampers the interpretation of the data, and does not justify our statement “shows a clear correlation with snow depth”. In response to this comment and other comments (also by reviewers #2 and #3), we replaced the original Figure S2. The latter figure showed area-averaged backscatter values, and no in situ measurements, because only a few were available above 2500 m elevation in the European Alps. The revised figure (Figure S3) shows time series of backscatter in co- and cross-polarization and polarization ratio for 4 representative individual sites, along with in situ snow depth measurements. This new figure clearly demonstrates the correlation between the cross-pol ratio and snow depth, and also gives a clear view on the general behaviour of backscatter over time, as was intended with the original Figure S2.

6. P19, line 419. You state that applying the ratio “may partially mitigate impacts from temporal changes”. I assume you tried also correlating only the cross pol? And apparently, the cross-co pol ratio yielded better results? This should be explicitly stated, now the justification for using ratio in place of just the cross pol backscatter is not very robust.

R.1, C.6: The better performance of the ratio (cross-pol over co-pol backscatter) compared to the cross-pol backscatter is now clearly visualized in the (revised) Figure S3. We rephrased the manuscript text on P.22, L.472 as follows: **“The cross-polarization ratio $\sigma_{vh}^0/\sigma_{vv}^0$ (in linear scale, converted to dB) shows overall a stronger correlation with snow depth than σ_{vh}^0 (Fig. S3, middle and right columns). Taking the ratio may partially eliminate the effects of temporal changes in the ground surface, vegetation, or snow conditions, which similarly impact both co- and cross-polarization.”**

7. P20, lines 426-439. It is not entirely clear to me whether you applied both am and pm orbits in your analysis. I am assuming both, but would it not be more prudent to apply only am orbits, in particular in the light of you discussion here? You would lose some temporal resolution, but have less chance of wet snow? Or, did you consider applying a wet snow mask, based on mapping wet snow with S1 itself, following e.g. temporal change detection methods presented by Nagler et al. (reference 23 in the manuscript)? In the least, these methods could be commented as they could be highly complementary to your snow depth retrievals. In this context I am also

not sure of what is meant by the last sentence in the paragraph (lines 438-439): what do you mean by wet snow observations being “evaluated throughout the paper”? Do you mean to say that in practice, you included all data, with no regard to possible wet snow?

R.1, C.7: Both ascending (6 pm) and descending (6 am) orbits were used. This is now clarified by a new Figure S2, showing a flowchart of the processing, and by the added text on P.19, L.391: **“Four sub-sets of Sentinel-1 data (i.e., ascending (6 pm) and descending (6 am) data from Sentinel-1A and -1B) were pre-processed separately and combined into a single Sentinel-1 dataset”**

Using only the 6 am (descending) overpasses could partially mitigate impacts from wet snow on the snow depth retrievals, at the expense of a reduced temporal coverage. But more importantly, several areas in the Northern Hemisphere are only observed by either ascending or descending overpasses. Therefore, the restriction to 6 am Sentinel-1 measurements would cause large spatial gaps in the retrievals.

The potential of Sentinel-1 to derive the wet/dry-state of snow was mentioned in the original manuscript on P.24, L.532 (P.x, L.x here refers to the revised manuscript): “In fact, previous studies demonstrated the retrieval of the snow wet-dry state²³ **(and even liquid water content⁵⁴)** directly from C-band SAR observations. **Wet snow maps derived from Sentinel-1 could be provided as auxiliary information (quality flag) with the SD observations”**.

With “wet snow observations being evaluated throughout the paper”, we indeed wanted to say that we included all data, irrespective to its wet/dry-state. We rephrased our statement in the manuscript at P.22, L.488: **“The primary objective of this study is to map the snow depth for dry snow conditions. However, we did not exclude wet snow conditions from the analysis and assess the performance of the retrievals throughout the snow season”**.

8. P21 lines 454-459. Fine for using the basic index modifier in eq (3) related to vegetation; applying exactly the same parameter b as for passive microwave, however, would require some further justification. This is because the observation geometries and wavelengths between S1 and passive microwave sensors are very much different. Did you experiment adjusting parameter b ? While optimizing both a and b simultaneously might not yield satisfactory results, taking forested grid cells with a range in FC, but more or less equal SD (+/- 5 cm, for example) from your training dataset, and attempting to optimize b might provide another optimal value?

R.1, C.8: We have tested different values of b and found the value of 0.6 (as used for passive microwave) to perform well. For instance, the spatial correlation of 0.76 (“Overall R_s ” in Table S2) for February 2018 would decrease to 0.69 if no forest correction would be applied (i.e., $b = 0$). That is, while the b -value of 0.6 may be sub-optimal, the impact of the forest correction is secondary.

We did not formally optimize the value of b because this would require splitting the in situ measurements into separate calibration and validation sub-samples. In some mountain ranges, this would reduce the number of sites below what would still be acceptable for use in the validation. To acknowledge the fact that we did not optimize the value of b , we added the following text to the manuscript on P.23, L.508: **“The performance of the algorithm can potentially be further improved by optimization of b ”**.

9. P21 line 469-P22 line 478. As the authors point out, experimental data making use of cross-polarized C-band data should be collected to study the effect of wet snow; also it would be necessary to further corroborate the claims presented in the paper, and notably to increase understanding on the cross-polarized response at

different wavelengths. This could be made more clear. In this regard, authors could perhaps state more explicitly that such data has been collected in the past, notably by Kendra and others as well as Strozzi and others – both references can be found in the paper. In particular, the authors should highlight (briefly) the main findings of those studies, and why they do not fulfill the requirements needed to corroborate the findings here. The study using artificially accumulated snow by Kendra and others did, after all, find indications of increasing cross-polarized backscatter at C-band, while measurements by Strozzi and others did not (possibly due to a high level of ground backscatter). Both experiments were rather short in time and/or limited to a few sites, and thus limited in representing natural snow in different stages of metamorphism.

R.1, C.9: We agree and have added text on P.21, L.444. The text is quoted below in R.2, C.1.

3. Reviewer #2 comments

The authors find that C-band cross-pol SAR contains information about snow accumulation in mountains. They present an algorithm and calibrate global parameter values between in situ data and SAR. The validation demonstrates that the retrievals have adequate skill to be useful in mountain areas.

This timely, important finding should be published; it is of broad interest, with the mere fact of it working appealing to remote sensing of snow community, and the broader perspective of measuring snow in global mountains appealing to a wider audience. The manuscript is generally well-written, and the analysis is sound. All of the edits I'm asking for relate to better explaining the measurement principle in the context of existing theory and observations which have not anticipated the result shown here. Honestly, I think the paper would be better if the authors said that it is not yet clear why C-band cross-pol is sensitive to such deep snow, but that this paper provides empirical evidence that this method works.

We would like to thank the reviewer for providing encouraging and constructive feedback to our work. We fully agree with the reviewer that we do not yet understand every aspect of the C-band cross-pol sensitivity to snow, but that we do provide ample evidence that the empirical retrieval algorithm works in a consistent way. In the revised version, we attempted to strike a balance between the justification and explanation of the theory (see for instance responses R.2, C.1-2) and the verification against in situ measurements, e.g. for deep snow (R.2, C.7). Please find below a detailed point-by-point reply to the comments.

Specific Comments

1. Two in situ observation studies are referenced in explaining the measurement principle references 19, and 26, King et al 2005 and Kendra et al 1998, respectively. Kendra et al. find that C-band cross-pol backscatter (see their Fig 7a) seems to saturate at about 50 cm, which is inadequate in the mountains. How do the authors fit their result with the much lower saturation depth observed by Kendra et al 1998? A third paper deserves mention: Chang et al. 2014. They look at X- and Ku- data from NoSREx, an important in situ radar snow experiment in Finland. They document cross-pol data with some response to snow depth at Ku-band, but almost no response at X-band. Why wouldn't the expectation be that there is even less response at C-band?

R.2, C.1: The few studies we found in literature that discuss C-band cross-polarized backscatter in view of snow depth (or snow mass) are using ground-based instruments and are fundamentally contradicting each other. As the reviewer explained in the above comment, Kendra et al. (1998) found a clear increase in cross-pol backscatter (~7 dB) for a snow depth increase from 20 to 60 cm, and only a minor further increase from 60 to 102 cm. This would suggest that (1) C-band cross-pol backscatter is clearly sensitive to snow depth, and

(2) the signal saturates around 60 cm. In sharp contrast, Strozzi and Mätzler (1998) found that C-band cross-pol backscatter slightly decreases with increasing snow depth. Strozzi et al. (1997) briefly discuss the contradicting results with Kendra et al. (1995; an earlier reporting of the 1998 paper), and argue that the experiments by Kendra took place on smooth, bare ground, with very low backscattering in snow-free conditions. They stated that the snow-free backscatter in their study was substantially larger (by 10 dB), through which the snow volume scattering contribution could no longer be noticeable. Strozzi and Mätzler (1998) also highlighted the need to compare their ground-based remote sensing results with satellite-based backscatter measurements. Our study analyses for the first time satellite-based backscatter measurements across the entire Northern Hemisphere mountainous area and agrees with the findings of Kendra that cross-pol backscatter increases with snow depth (see revised Figure S3). Moreover, the cross-pol backscatter values obtained by Sentinel-1 for snow-free conditions are typically larger than -20 dB (see for instance Figure S3), and larger than the values (mostly below -20 dB) reported in their Figure 3 by Strozzi and Mätzler (1998), therefore not supporting their statement about the volume scattering contribution.

A potential reason why Strozzi and Mätzler (1998) did not find an increase in backscatter with snow depth could be related to the design of the radar backscatter measurements. In short, the backscatter measurements were calculated by integrating the contributions only from a few (they stated “usually no more than 3”) dominant “surface” scattering layers, such as the ground-snow interface, the air-snow interface, and layer interfaces within the snowpack. As acknowledged by the authors, this type of backscatter calculation does not include multiple scattering, which is potentially important (e.g., Chang et al., 2014; King et al., 2005).

There could be several reasons why Kendra et al. (1998) observed saturation at 60 cm, while our results show no signs of saturation for snow depths greater than 2 m. The experiments by Kendra et al. (1998) used artificial snow, which in some of its characteristics deviated from natural snow. For instance, the shape of the artificial snow grains was rounded, which will cause less of the anisotropic scattering that typically depolarizes the signal. The snow particles were reported to have a small diameter (0.27 mm) that may also reduce scattering. The vertical snow profile was homogeneous (temperatures were below freezing during the 6-day experiment), whereas depolarization is typically caused by inhomogeneities. Finally, their scatterometer system had a footprint of 18 m × 30 m, with more spatially homogeneous snow conditions compared to those within the coarser 1-km² grid cells in our study.

The analysis by Chang et al. (2014) distinguishes between two types of radar measurements, i.e., tower-mounted scatterometer (SnowScat) and airborne SAR (SnowSAR). The authors ran model simulations with DMRT-QCA and DMRT-Bicontinuous, which offer a more realistic description of scattering that departs from Rayleigh theory (in which scattering is proportional to the fourth power of frequency and third power of grain size) and accounts for scattering from clusters of snow particles rather than calculating (and super-imposing) the scattering from each individual snow particle independently. Contrasting results were obtained with the two types of radar measurements. The SnowScat measurements showed good sensitivity to snow depth at Ku-band, but not at X-band, whereas the SnowSAR measurements showed very similar sensitivities between both frequencies. The DMRT model simulations were able to simulate each of these cases, including the sensitivity to snow depth of X-band SnowSAR data.

To summarize, the above-mentioned studies suggest a few important aspects: (1) tower-mounted radar measurements can show fundamentally contradicting results, such as either the increase or decrease of backscatter with increase of snow depth. (2) tower-measurements do not always match what is observed from airborne and satellite (e.g. Sentinel-1) data, despite often being deployed for the testing of satellite concepts. Potential reasons for discrepancies in (1) and (2) include the radar backscatter measurement principle, the smaller spatial scale, as well as site-specific snow, vegetation and ground conditions. (3) Snow scattering is

likely less strongly dependent on the proportionality of frequency versus grain size than was long assumed based on Rayleigh theory. This is now increasingly supported by state-of-the-art, physically-based radiative transfer theory, which acknowledges the potential importance of scattering on conglomerates of snow, rather than individual particles (see also our reply to R.2, C.2 below).

The following text was added on P.21, L.444: “**The temporal variations in σ_{vh}^0 with snow depth evolution have, to our knowledge, not been investigated before with C-band satellite observations. Only few studies deployed tower-mounted radar instruments, with contradictory results. Overall, our study aligns well with the scatterometer measurements over a site in Michigan, USA, revealing an increase in σ_{vh}^0 with an increase in snow depth²⁶. However, the scatterometer measurements showed signs of saturation for depths exceeding ~60 cm, which is not observed in the Sentinel-1 observations. Potential causes for this discrepancy include the use of artificial snow in the scatterometer experiments (characterized by a homogeneous layer of snow composed of small, rounded particles and the absence of snow melt-freeze metamorphism), or differences in the spatial support of the measurements (18 m × 30 m for the scatterometer versus 1 km² for the processed Sentinel-1 observations). In strong contrast, a (minor) decrease in σ_{vh}^0 with increasing snow depth was observed from scatterometer measurements over a site in the Swiss Alps⁵⁰. An inverse relationship can occur with site-specific ground, vegetation and snow conditions, if the attenuation of ground scattering by the snowpack is stronger than the scattering contribution from the snowpack²⁵. However, the results of the study in the Swiss Alps could also be impacted by the backscatter measurement principle: the total backscatter was calculated by integrating the scattering contributions from a few dominant surfaces, i.e., the snow surface, the ground surface and/or horizontal layers within the snowpack, and was thus not including the multiple scattering⁵⁰.**

In agreement with the scatterometer measurements in Michigan²⁶ and with our Sentinel-1 observations, radiative transfer model simulations generally indicate an increase in σ_{vh}^0 with snow depth. Recent model developments tend towards a relatively weak dependence on frequency⁴⁸. A critical aspect in this context is the development of state-of-the-art, theoretically-based radiative transfer models, which allow for simulating volume scattering from snow, represented by clustered, non-spherical particles^{17,47-49}. This presents a major improvement over conventional solutions for scattering from individual, spherical snow particles using Rayleigh theory, assuming that scattering is proportional to the fourth power of frequency and third power of grain size⁴⁹.”

2. Most theory does not adequately treat cross-pol: e.g. Shi & Dozier (2000) (ref 25 in the submission) state that their approach for analyzing Sir-C only handles co-pol as it only treats first order back-scattering. I think it would be nice to make this point in the manuscript. Models capable of simulating cross-pol should also be mentioned. Xu et al. (2012) use a more sophisticated model and show a decreasing cross-pol response to snow depth as frequency decreases from 17 to 10 GHz; while they do not show C-band results, the cross-pol response in that case would be nearly zero by extension. Du et al. 2010 and Yueh et al. 2009 hypothesized that (higher-frequency) cross-pol response was likely due to grain shape asymmetry. I think that pointing to these hypotheses as to why the cross-pol response is observed should be added.

R.2, C.2: Scattering models have greatly improved after the development of DMRT, including approaches that describe the snow medium as clusters of particles with non-spherical shape, such as the bicontinuous medium version (e.g., used by Xu et al., 2012). The cross-pol backscatter simulations by Xu et al. (“hv-tot” in their Figure 17) show a very similar ~2.5 dB increase at both Ku-band and X-band, for an increase in snow depth from 10 to 80 cm. Thereby, the Ku-band backscatter seems to saturate around 60 cm snow depth, whereas no (or less) saturation is observed for X-band. Unfortunately, phase matrices for the bicontinuous DMRT have

not yet been calculated at C-band. We are currently initiating collaboration with Dr. Shurun Tan to extend the bicontinuous DMRT for C-band, and plan to test the simulation of the Sentinel-1 backscatter in future research.

The studies by Xu et al. (2012) and Du et al. (2010) have now been integrated into the manuscript (see also the response R.2, C.1 above). The studies by Shi and Dozier (2000) and Yueh et al. (2009) were already included.

3. Line 28-29: I think this overstates the finding. Sentinel-1 C-band cross-pol is clearly correlated with snow depth. But I think the abstract, while it has to be brief, ought to hint that this is a purely empirical approach and that the physics behind why it works are not well understood.

R.2, C.3: To not overstate the finding and indicate the empirical nature of the approach, we have revised the sentence in the abstract on P.2, L30 to: **“Here, we demonstrate the ability of the Sentinel-1 mission to map the snow depth in the Northern Hemisphere mountains at 1 km² resolution using an empirical change detection approach”**.

To further highlight the empirical aspect of the approach, we also added the following text:

- P.4, L.85: “We use this ratio **in an empirical change detection algorithm** (see supplementary section Retrieval algorithm **and Fig. S2**)”
- P.7, L.181 (conclusion): **“We provide clear evidence of the value of the empirical Sentinel-1 retrievals through a comparison against point-scale measurements at ~4,000 sites and coarse-scale global reanalysis data”**.
- P.23, L.493: “The snow depth (SD; in m) retrieval algorithm relies on **an empirical** change detection method applied to the Sentinel-1 measurements of the cross-polarization ratio ($\sigma_{vh}^0/\sigma_{vv}^0$; in dB)”.

4. Line 66-67: I think you should mention here that the method relies on calibrating the radar response.

R.2, C.4: The method for the snow depth retrievals is discussed in detail in the supplementary. We believe that highlighting a single aspect of it in the main article may cause confusion and prefer that the reader interested in the processing refers to the detailed text in the supplementary.

5. Line 79: ref 19 by King does not show cross-pol response > co-pol response at Ku band, in my understanding.

R.2, C.5: We provide the reference to King et al. (2015) here because this study presents a clear discussion of mechanisms that contribute to cross-pol backscatter (albeit at a higher frequency). Note that the sentence in question does not talk about the relative magnitudes of the cross-pol and co-pol responses.

6. Line 88: I think again you need to say that you are calibrating to in situ data, and that the measurement principle is not yet well understood.

R.2, C.6: Please refer to our answer on comment #4 above. In our opinion, it is not necessary to emphasize the rescaling with parameter a . The latter is constant over both space and time (P.23, L.503), and does not impact the correlation metrics presented in the validation (it only impacts the MAE and bias).

7. Line 92-102: In this section, I personally would think that showing a few select sites with in situ data in the supplement or even the main paper would be far more convincing than what is shown, which are the radar data

averaged over large regions in Fig S2, and R values in Fig S4. This to me would go a long ways to showing that the signal is responsive to deep snow.

R.2, C.7: We have replaced Figure S2 with the new Figure S3, showing time series of backscatter in multiple polarizations for 4 different sites, with corresponding in situ snow depth measurements. We agree with the reviewer that this presents a more convincing proof that the signal is responsive to deep snow. See also response R.1, C.5.

8. Line 179-180: It is not clear why the observations are sensitive to deep snow. I would reword.

R.2, C.8: We are confident that this is now demonstrated by the new Figure S3 and the corresponding discussion.

References

Chang et al., 2014. Dense Media Radiative Transfer Applied to SnowScat and SnowSAR, IEEE JOURNAL OF SELECTED TOPICS IN APPLIED EARTH OBSERVATIONS AND REMOTE SENSING, VOL. 7, NO. 9, SEPTEMBER 2014.

Du et al. 2010. Comparison between a multi-scattering and multi-layer snow scattering model and its parameterized snow backscattering model, Remote Sensing of Environment 114 (2010) 1089 – 1098.

Xu et al. 2012. Electromagnetic Models of Co/Cross Polarization of Bicontinuous/DMRT in Radar Remote Sensing of Terrestrial Snow at X- and Ku-band for CoReH2O and SCLP Applications. IEEE JOURNAL OF SELECTED TOPICS IN APPLIED EARTH OBSERVATIONS AND REMOTE SENSING, VOL. 5, NO. 3, JUNE 2012

Yueh et al. 2009 Airborne Ku-Band Polarimetric Radar Remote Sensing of Terrestrial Snow Cover. IEEE TRANSACTIONS ON GEOSCIENCE AND REMOTE SENSING, VOL. 47, NO. 10, OCTOBER 2009

4. Reviewer #3 comments

My comments are in the attached pdf file. It is an original and very interesting manuscript that will interest a large audience but it brings more questions than answers. The authors need to convince me that the proposed algorithm is deriving the snow depth in mountainous areas. I look forward to read their reactions to my comments.

Comments inserted from pdf file: 3_reviewer_attachment_1_1558287321_convrt.pdf

We would like to thank the reviewer for providing constructive feedback to our work. In the revised manuscript, we now provide stronger support to the main claim of the paper that the proposed algorithm retrieves snow depth in mountainous areas (see for instance responses R.1, C.5; R.2, C.7; R.3, C.17 and the revised Figure S3). Please find below a detailed point-by-point reply to your comments.

References:

Studies has been done in some mountainous areas, Alpes, Rockies Canadian and Libanon. However, the snow wetness was a problem.

CORBANE*, C., SOMMA, J., BERNIER, M., FORTIN, J.P., GAUTHIER, Y*., DEDIEU, J.P. (2005). Estimation de l'équivalent en eau du couvert nival en montagne libanaise à partir des images RADARSAT-1. Conférence Snow Hydrology of Mediterranean Regions à Beyrouth, Liban, 15-17 décembre 2002, Journal des Sciences Hydrologiques, 50(2) : 355-370.

J.-P. Dedieu, N. Besic, G. Vasile, J. Mathieu, Y. Durand and F. Gottardi, Dry snow analysis in alpine regions using RADARSAT-2 full polarimetry data. Comparison with in situ measurements. IEEE International Geoscience and Remote Sensing Symposium (IGARSS'14), July 13–18, Quebec, Canada, pp. 3658–3661, 2014.

Arnab Muhuri, Surendar Manickam, Avik Bhattacharya, Snehmani, "Snow Cover Mapping Using Polarization Fraction Variation With Temporal RADARSAT-2 C-Band Full-Polarimetric SAR Data Over the Indian Himalayas", Selected Topics in Applied Earth Observations and Remote Sensing IEEE Journal of, vol. 11, no. 7, pp. 2192-2209, 2018.

R.3, C.1: We have investigated the above-mentioned references. For our work, we chose to present the snow depth retrieval results throughout the entire snow season, including both dry and wet snow conditions. That allows us to assess the performance under different conditions, including the impact of snow wetness. As clearly mentioned in the paper (P.22, L.488), the primary objective is to retrieve accurate snow depth for dry conditions, as we are aware that snow wetness may cause absorption of the radar signal and may thus impact the retrieval (see text on P.22, L478). To further clarify this, we edited the manuscript (see R.1, C.7).

In future research, we plan to (1) investigate into detail (through field campaigns) the impact of wetness on the snow depth retrieval, (2) assess the retrieval of the wet/dry-state of snow from Sentinel-1 (similar to Nagler et al., 2016), and (3) include the presence of wet snow as a quality flag in the retrieval product.

Nagler, T., Roth, H., Ripper, E., Bippus, G. & Hetzenecker, M. Advancements for snowmelt monitoring by means of Sentinel-1 SAR. *Remote Sens.* **8**, 348 (2016)

Also, pertinent references are missing:

ALGOSNOW - Contract No. 4000103180/11/NL/CT: Algorithms for Snow and Land Ice Retrieval using SAR data. ESA Report, 2013.

Rott, H. et al. "Development of snow retrieval algorithms for CoReH2O – Final Report, ESAESTEC Contract 22830/09/NL/JC, 2011.

Macelloni G, Brogioni M, Montomoli F, Fontanelli G. 2012. Effect of forests on the retrieval of snow parameters from backscatter measurements. *Eur J Remote Sensing*, 45: 121–132, 2012.

R.3, C.2: We added the reference to Macelloni et al. (2012) to the paper, on P.23, L.504: "Forests typically attenuate snow backscatter⁶² ...". Unfortunately, we do not have access to the ESA reports for ALGOSNOW and CoReH2O.

Figures

Figure 1: There are glaciers in the Mountainous areas. Those areas should be localized. It seems that the areas identified as "Mountainous, no snow" in Figure 1 included the glaciers.

R.3, C.3: Thanks for the suggestion. We have now processed glacier data from the Randolph Glacier Inventory 6.0 to localize glaciated areas and assess their impact on the snow depth retrieval. A figure showing the location of the glaciers in the European Alps, on top of Sentinel-1 snow depth, is shown in the reply R.1, C.2. The impact of glaciers on the calculated performance metric was found to be negligible, since only very few in situ measurements were located in pixels that were (at least partially) covered by glaciers (difference in spatial correlation below 0.001). We revised Figure 3 and removed glaciated areas from the area-wide Sentinel-1 snow volume estimates. There, for some mountain ranges (e.g., the Kenai Mountains, Chugach Mountains, and Saint Elias Mountains), a considerable impact was observed, since large areas within these ranges are glaciated. Finally, we included glaciated areas as a flag in the snow depth retrievals, so that potential users can decide whether or not to include glaciated areas, and could also evaluate the performance over these areas.

Figure 3: I don't understand the message in this Figure. What is the difference between Crossmarked Sentinel and Areas Wide Sentinel?

R.3, C.4: The cross-masking of Sentinel-1 is now better explained on P.7, L.164: “**the cross-masked Sentinel-1 observations (i.e., averaged only over grid cells that include measurement sites; purple dots in Fig. 3)**”, and in the Figure 3 caption: “**Bars represent the area-wide Sentinel-1 observations averaged over the entire mountain range, black crosses the average of in situ measurements, and purple dots the corresponding average of cross-masked Sentinel-1 observations, averaged only over grid cells that include measurement sites**”.

Sentinel-1 data processing:

Line 375-377: I did not understand the final processing done after the incidence angle normalization? What were the range of those incidence angles?

R.3, C.5: The final backscatter processing steps are the removal of outliers and the merging of Sentinel-1 ascending and descending data per day. This is now clarified on P.19, L.390: “**Outliers were removed by excluding values that are 3 dB above the 90th-percentile or 3 dB below the 10th-percentile of the time series. Four sub-sets of Sentinel-1 data (i.e., ascending (6 pm) and descending (6 am) data from Sentinel-1A and -1B) were pre-processed separately and combined into a single Sentinel-1 dataset**”. The incidence angle (assuming a flat surface) of Sentinel-1 ranges from 29.1° to 46.0°. The local incidence angle is calculated using the SRTM digital elevation model, and is more variable (from 0 to 90° in (a few) extreme cases; values outside this range were removed). The range in incidence angle has been added to the text on P.18, L.380: “the Sentinel-1 observations from the different orbits within one repeat cycle have different incidence angles, **ranging from 29.1° to 46.0° relative to a flat surface**”.

A schema would help to understand all the steps of the image processing. Those done by ESACopernicus (format of the images downloaded), those done by the authors.

R.3, C.6: We agree and have added a new flowchart (Figure S2) to illustrate the steps in the backscatter processing and snow depth retrieval. The standard processing techniques (box **a**) are performed in Google Earth Engine. All further steps (boxes **b** and **c**) are done by the authors. The processing now also runs fully automated, i.e., the snow depth retrievals are calculated daily, with a latency of 7 days.

Algorithm:

Line 450: Equation 3 shown that you take care of the evergreen forest cover fraction (in %) in a given pixel of 1 km, using an attenuation constant (b). However, elsewhere in the manuscript, it is indicated that you applied the algorithm above the tree line (2500 m).

R.3, C.7: We applied the snow depth retrieval algorithm to the full study domain, i.e. the mountain areas in the Northern Hemisphere, thus including forested areas. The reviewer is referring to Figure S2 of the original manuscript, which was showing backscatter (not snow depth) for pixels above the tree-line in the European Alps. The purpose of this figure was to show the typical backscatter response to snow in different polarizations over time, while trying to exclude impacts of vegetation/forests in the visualisation, by focusing on the area above tree-line (> ~2500 m). This figure is now replaced by a new Figure S3.

I would be curious to see how your Snow Indice values (equation 2) are correlated with the Snow depth? The correlation could be better than SD for pixels above the tree line.

R.3, C.8: Our study distinguishes between two correlation values: the temporal correlation and the spatial correlation. The temporal correlation of the snow index (SI) with the snow depth measurements is exactly the same as the temporal correlation of the snow depth (SD) retrievals with the measurements. We apply a multiplicative rescaling of SI with the parameter a that is constant ($a = 1.1$) in time (and space) and does not impact the temporal correlation. The spatial correlation is impacted by the rescaling from SI to SD through the use of a spatially-varying forest cover fraction (but not by the multiplicative rescaling with the parameter a). The impact of the rescaling based on forest cover is positive, but it is not of primary importance. Without this rescaling, the spatial correlation R_s (for February 2018) equals 0.69. With rescaling, this increases to 0.76 (Table S2). In the absence of forest cover, Equation 3 reduces to $SD = a SI$, meaning there is no influence of the rescaling on neither temporal nor spatial correlations.

Line 456-458: "Finally, the SD observations based on Eq. (3) are smoothed to further reduce the impact of Sentinel-1 observation noise. This is done using linear inverse distance weighting with a 2 km radius in space and a 10 day radius in time".

I don't agree that there are still a noise in the Sentinel-1 data after all the smoothing done in the processing of the data (filtering and resampling to 1km). As shown in Figure S2b, there are no variation in the VV signal during the summer (-8 dB) and one would expect that raining and drying period would affect the backscattering signal from the ground (soil- vegetation) unless the values shown are for rocky area. Also, it would be expect that the soil freezing (at least in the beginning of the winter when the snow cover is shallow) a decreasing in backscattering will be recorded in VV polarization but the signal in winter is also stable at -8 dB.

R.3, C.9: Noise can still persist in backscatter averaged over relatively large spatial scales because the backscatter intensity distribution typically has very long tails (i.e., very low or high extreme values). Besides further filtering noise, the smoothing also reduces short-term, high-magnitude fluctuations in the snow depth retrievals. Passive microwave retrievals apply similar smoothing for the same reasons (Kelly et al., 2003; reference 10 in the manuscript). This has been added on P.23, L.513: "Finally, the SD observations based on Eq. (3) are smoothed to further reduce the impact of Sentinel-1 observation noise **and to reduce short-term, high-magnitude fluctuations**¹⁰".

The original Figure S2b focused on areas at high elevation, indeed mostly associated with rocky surfaces. The motivation to focus on these areas was to eliminate as much as possible the impacts on backscatter from soil

and vegetation conditions, to visualize better the impact of snow. Note that there may still be some impact of soil and (short) vegetation. For instance, VV backscatter decreases by about 0.5-1 dB around the end of September, likely caused by decreasing vegetation or moisture, or soil freezing. However, the effects for individual pixels are “blurred” in a density plot if they occur at different times, which was likely the case.

We have revised Figure S2 (now Figure S3) by showing time series for 4 individual pixels with corresponding snow depth measurements, in different mountain ranges. This avoids some of the issues mentioned above (e.g., the loss of information by averaging many pixels). See also response R.1, C.5.

Line 464: For the optimization of parameter a , you excluded the time periods from March to August to avoid impacts from wet snow. However, it seems that you applied equation 2 and 3 to Sentinel data acquired in the same period (Figure S5)?

R.3, C.10: As mentioned previously, our primary objective is to provide accurate snow depth for dry snow conditions. In the future, we plan to flag snow depth retrievals if wet snow conditions are suspected. To have the most accurate results for dry snow conditions, the rescaling parameter should be based on primarily dry snow conditions as well, which is why we excluded the months of March to August during optimization.

Line 470-474: To further improve the retrieval algorithm, your recommend future research to investigate the impacts of snow wetness on $\sigma_0^{vh}/\sigma_0^{vv}$, for instance using tower-mounted radar measurements and including local observations of snow liquid water content and snow density in the in the retrieval algorithm. I don't think that you can hope to get better results in including more parameters that vary in spatially and temporally like snow liquid water content and snow density. Those information will be difficult to get globally as they are already scarce locally. Further, as shown in Figure S2d, in 2017, between March 1st and June 1st, the range of the Ratio is only 2 dB.

R.3, C.11: As stated in the corresponding sentence, “we recommend future research to investigate” these opportunities for improvement. We do not claim their success, but we would like to highlight that AMSR-2 passive microwave snow depth retrievals make use of a grain size and snow density evolution model (Kelly et al., 2003) which is very beneficial for the quality of the retrievals (personal communication with Richard Kelly, University of Waterloo, Canada). A similar approach could be followed for Sentinel-1 and is expected to improve the performance. Furthermore, as mentioned in the paper on P.6, L.149, snow density estimates derived from snow climate classes are available at the global scale through Sturm et al. (2010).

Kelly, R. E., Chang, A. T., Tsang, L. & Foster, J. L. A prototype AMSR-E global snow area and snow depth algorithm. *IEEE Trans. Geosci. Remote Sens.* **41**, 230-242 (2003).

Sturm, M. et al. Estimating snow water equivalent using snow depth data and climate classes. *J. Hydrometeorol.* **11**, 1380-1394 (2010).

Then, I am not comfortable with you recommendation (line 470-474) which is in contradiction with one of your statement (line 420) that taking the ratio may partially mitigate the impacts from temporal changes in the ground surface, vegetation, or snow microstructure conditions. From my experience, the ratio is mainly reducing the slope effects in Mountain Area and also the variation in local incidence angles which are link and then vegetation and snow.

R.3, C.12: The improvement in correlation with snow depth when taking the polarization ratio is now clearly visible in the revised Figure S3. This figure also clearly shows for instance a similar decrease in VV- and VH-

polarized backscatter in autumn (caused by changes in soil and vegetation conditions), which reduces when using the ratio. The impact of the slope is (at least partially) corrected by using a change detection technique (a strong impact of slope is observed in a map of backscatter on a single day, but the change in backscatter relative to a previous day shows a much reduced impact). If the ratio would further reduce the slope (and incidence angle) impact, this would only make a stronger case for using the ratio. We do not agree with the reviewer that the statements in line 470-474 and 420 in the original manuscript (or P.24, L.527-533 and P.22, L.473 in the revised manuscript) are in contradiction. Reducing the impacts of certain snow conditions (for instance snow density or wetness) does not mean that including auxiliary information can no longer improve the results.

Regarding, the availability of the snow measurements in VH and VV polarisation for different frequencies, some data are available in some recent ESA reports about snow algorithms for SWE and the NASA SnowEX.

R.3, C.13: We added the following references to studies investigating the (airborne) backscatter response in X- and Ku-band: Du et al. (2010), Xu et al., (2012), and Chang et al. (2014). The C-band studies, to our knowledge, were focused on the use of co-pol backscatter, the retrieval of wet-dry state of snow, or applications of radar polarimetry (target decomposition). The former two are in our opinion sufficiently addressed in the paper, while the latter is less relevant for Sentinel-1, which does not measure the required full scattering matrix. Co-authors of our paper (E. Kim, L. Brucker, and H.-P. Marshall) have organized the NASA SnowEx 2017 campaign (H. Lievens was participating). During SnowEx 2017, no airborne observations were collected in C-band. In future SnowEx years, our team will collect tower-based C-band radar measurements at field sites in Idaho and Colorado, USA.

Du, J., Shi, J. & Rott, H. Comparison between a multi-scattering and multi-layer snow scattering model and its parameterized snow backscattering model. *Remote Sens. Environ.* 114, 1089-1098 (2010).

Xu, X., Tsang, L. & Yueh, S. Electromagnetic models of co/cross polarization of Bicontinuous/DMRT in radar remote sensing of terrestrial snow at X- and Ku-band for CoReH2O and SCLP applications. *IEEE J. Sel. Top. Appl. Earth Obs. Remote Sens.* 5, 1024-1032 (2012).

Chang, W., Tan, S., Lemmetyinen, J. & Tsang, L. Dense media radiative transfer applied to SnowScat and SnowSAR. *IEEE J. Sel. Top. Appl. Earth Obs. Remote Sens.* 7, 3811-3825 (2014).

Line 494-498: I agree with the authors that ideally, the validation of the Sentinel-1 observations would therefore be performed using data that can be processed at the matching scale, such as gridded estimates from airborne lidar, regional model simulations or high-resolution model reanalysis but that such information is not available at the global scale. Then, the authors should be more cautious in their conclusions regarding the correlation between the Sentinel-1 prediction model (equation 2 and 3) and the measured Snow Depth as well in the interpretation of Figure S5.

R.3, C.14: Owing to measurement and representativeness errors in the (point-scale) in situ measurements, the presented correlation metrics are conservative estimates of the actual retrieval performance. The metrics are likely to improve when using validation data at the matching scale. To clarify this, we added the following text on P.25, L.551: **“We use the point-scale in situ measurements for evaluating the performance of the Sentinel-1 snow depth retrievals. However, this likely provides a conservative estimate of the true retrieval performance”**.

Figure S5 presents temporal snow depth retrievals averaged over several sites, and should therefore be less prone to representativeness differences that affect the comparison with individual in situ site measurements.

In particular for the snow melt period (March 1st- June 1st) , the decrease in backscattering is link on the wet snow absorption of the signal in both VH and VV polarisation and that absorption increases with the warming of the air temperature and increases on the liquid water content at the air/snow interface. This decreasing of the snow signal correlated indirectly with the melting of the snow cover and the decreasing of the snow depth. Then, I don't agree with your interpretation (line 420-424) that the ratio decreases because of the relatively higher decrease rate of σ_{vh} , caused by the reduced volume scattering (and depolarization) in an increasingly shallow snowpack.

R.3, C.15: During snow ablation, it is clear from the data (e.g. Figure S3) that the cross-pol backscatter decreases more than the co-pol backscatter, and that the cross-pol ratio usually remains higher than that of snow-free conditions. Nevertheless, we have deleted “caused by the reduced volume scattering (and depolarization) in an increasingly shallow snowpack” from lines 420-424 (in the original manuscript), because we agree with the reviewer that, besides remaining snow volume scattering, also other mechanisms could have an impact. We therefore merged the corresponding paragraph with the subsequent paragraph that provides a number of caveats on dry/wet snow conditions (P.22, L.478): “Changes in snow properties between dry and wet conditions, such as in snow microstructure and liquid water content, potentially modify the proportionality of volume versus surface scattering, and therefore the sensitivity of the C-band observations to snow depth. Thereby, the sensitivity to snow depth for wet snow is much more uncertain. Strong melt events, with associated high liquid water contents, may cause fluctuations in $\sigma_{vh}^0/\sigma_{vv}^0$ as the signal is more strongly reflected and absorbed **by wet snow layers**⁵⁰. Although this effect may be less severe at C-band than at higher frequencies, and may occasionally be alleviated by the refreezing of snow⁵⁰ prior to the early-morning (6 am) and evening (6 pm) overpass times of Sentinel-1, it is likely to have a confounding impact on the C-band sensitivity to snow depth. Similarly, the backscatter signals are impacted by successive melt-refreeze cycles that can modify the microstructure and stratigraphy of the snowpack”.

As explained previously, our main concern remains the snow depth retrieval for dry snow conditions, as stated on P.22, L.488: “**The primary objective of this study is to map the snow depth for dry snow conditions. However, we did not exclude wet snow conditions from the analysis and assess the performance of the retrievals throughout the snow season**”.

Besides, the backscattering is increasing in early summer (approximately in June) in both polarisations with the decrease of the snow cover fraction (in %) within a 1km pixel and the increase contribution of the wet soil and the vegetation. Then, in the late snow season the VH/HH ratio don't vary much (Figure S2D). I am surprised that you derived SD values that correlate with the field data for that period (June-July), although it is difficult to see on the graphs of Figure S5 and the correlation shown are globally for the two years and not by season.

R.3, C.16: The increase of backscatter in early summer confirms previous studies at higher frequencies. For instance, Nghiem et al. (2001) provide a clear description of the temporal behaviour of backscatter over the snow season, relating the increase in early summer to the disappearance of wet snow (reducing the absorption) and simultaneous thaw and greening (increasing backscatter).

We agree with the reviewer that in some cases, there seem to be slightly lower cross-polarized backscatter values for high snow depths in the late snow season, often following a short period of melt. Figure S3b shows an example for which the snow depth around the end of April is higher than that of early March when the

maximum cross-pol backscatter is observed. In-between these two periods with high snow depths, there is a short period of melt. We still need to further investigate what is causing this behaviour. Reasons can be manifold. For instance: (1) The in situ measurements may not well represent the pixel. Many in situ measurements are taken in relatively flat areas where snow can persist for a longer time compared to nearby slopes. A fraction of the coarser-scale Sentinel-1 pixel may still contain wet snow conditions, particularly if the pixel comprises areas at lower elevation or on a south-facing slope. (2) Melt-freeze cycles may increase the impact of snow layering (incl. ice lenses), which may reflect horizontal-polarized waves stronger than vertical-polarized waves, thus reducing the cross-polarized backscatter. (3) Wet or mixed wet/dry snow can be buried by a fresh dry snow layer, but still absorb/reflect part of the signal. (4) Scattering could increase in VV-polarization (decreasing the VH/VV ratio), as metamorphism causes larger snow particles that may become more efficient scatterers also in co-polarization.

We will investigate the exact cause for this behaviour in future research. To this end, we will perform field campaigns during the next two winters in the US, collecting tower-mounted radar measurements in C-band and corresponding detailed snow observations.

Nghiem, S. V. & Tsai, W.-Y. Global snow cover monitoring with spaceborne Ku-band scatterometer. *IEEE Trans. Geosci. Remote Sens.* **39**, 2118-2134 (2001).

I am more comfortable with the link between the increase of the Ratio in winter with the increase in volume scattering from the snowpack in VH polarization where there is no coniferous tree and the snowpack is dry as this behavior has been reported elsewhere for C- and X-band. However, other environmental parameters (unfrozen ground) or images processing could explain the good correlation. May be the huge dataset used for the optimisation (a parameter) but limited to two winters could explain the good correlation and the relatively small RMS with field measurements which are also not always representative of the heterogeneity of the snowpack.

R.3, C.17: We are convinced that the correlation is not originating from the aspects mentioned by the reviewer:

- The scaling factor a is constant over time and space, and therefore does not at all impact the correlation.
- The scaling based on forest cover has a slightly positive impact, but as explained above, is of secondary importance.
- The applied processing steps are standard and are not influencing the correlation between backscatter and snow depth. This should now be clear from Figure S2.
- The insulating effect of snow on ground temperature may have an impact on backscatter, for instance for shallow snow in early winter. However, in many cases, snow cover causes ground temperature to remain stable for most of the winter period, during which snow accumulates and we observe an increase in the cross-polarized backscatter. If temperature would be the dominant influence, we should see a similar increase in the co-polarized backscatter, which is not observed. Further, the inter-annual variability in mid-winter backscatter correlates well to the inter-annual variability in snow depth (shown for instance in Figure 2 for the Alps and Sierra Nevada). Inter-annual ground temperature differences are not expected to be very different due to the insulation by the typically thick snowpack in these areas. Also, in several places (see for instance Figure S3d), we observe a cross-polarized backscatter in winter that is considerably higher than in summer, during unfrozen conditions. Finally, in bedrock pixels, such as those shown in the original Figure S2, ground temperature should not have a strong impact, while a considerable increase in cross-pol (and not in co-pol) was still observed. We are aware that previous studies have attributed backscatter changes at C-band in snow-covered areas to the thermal resistance (e.g., Bernier et al., 1999). But, it is important to consider that these studies were carried out in tundra environments with relatively

shallow snow, and were using only co-polarized (HH) backscatter, therefore presenting a very different case study.

Bernier, M. et al. Determination of snow water equivalent using RADARSAT SAR data in eastern Canada. *Hydrol. Proc.* **13**, 3041-3051 (1999).

In conclusion, the authors still need to convince me that their algorithm is estimating the snow depth in mountainous areas.

R.3, C.18: We sincerely hope that our modifications to the manuscript (in particular, Figures S2 and S3) and the responses above adequately addressed the reviewer's concerns. We are confident that the revised manuscript better supports the algorithm and the retrieval results of snow depth.

Reviewer #1 comments

The authors have addressed all my previous comments. As a result, I think the manuscript is almost ready for publication.

My remaining concern is with the new figure S3, which in my view gives a somewhat overly positive view of the correlation between the v_h/v_v backscatter ratio and snow depth, apparently due to different scales used in the figure axes for snow depth (in S3a the scale for snow depth is from 0 to 2 m, in S3b from 0 to 4 m, and in S3c&d from 0 to 3 m). Unless I am mistaken, this has been done to visually improve the match between v_h/v_v backscatter and SD in the figure? I think it would be better to either provide uniform scaling across all figures, and accept any possible visual bias that results. If the authors prefer to keep the present format, it would be good to make a note of the scaling in the text (e.g. around line 475 in the supplementary section) and in the figure caption.

Reviewer #2 comments

Generally the authors have responded well to my concerns raised on the earlier round of reviews. First, I want to say that I find Figure S3 very convincing. Thank you for adding this. Second, the addition of the word “empirical” I think is excellent, in response to my comment 3 on the previous round. And the additional discussion of empirical literature in the Supplementary section is very helpful.

I think there are two things that need to still be improved before publication, however.

First, related to my comment 4 on the previous round of review, you still have to go to the supplement to find that this is an approach based on calibration to a subset of in situ data. The authors' response to not including “calibration” language in the main text is that they don't want to include any of the methodology used to compute SWE in the main text in order to avoid confusing readers. I disagree: I think that telling readers that the approach relies on calibrating to a subset of available in situ data, and validating against the rest is both of great value to the readers, and very easy to understand. I think that readers of Nature Communications understand calibration-validation studies, as well as the difference between a measurement based on first principles versus one based on calibration. Please add the language to the main text.

Second, it is still not clear in the main article how shocking it is that C-band is sensitive to mountain snow, in the context of decades of tower-based, airborne, and modeling studies on the problem of using active or passive microwave measurements to measure snowpack. The language in lines 64-68 in the current manuscript is: “The use of C-band backscatter for estimating snow depth (or mass, related to depth by the density) has long been swept aside after earlier studies had shown a limited sensitivity [24,25]. However, these studies were mostly surveying shallow snow outside mountain environments and, more importantly, were limited to backscatter measurements in co-polarization.”

Lines 77-82 sort of make it sound like the authors' result is in agreement with previous studies. Lines 64-88 and 77-82 are just not adequate, in my view. From their paper and their response on the previous round of review, the authors have convinced me that previous literature and modeling studies have not shown that C-band cross-pol should not be sensitive to deep snow depth. However, I think they need to go a little further in the main manuscript. In my opinion, I think readers of the main manuscript need to know that in decades of work on remote sensing of snow, no one has actually published results on C-band cross-pol in deep snow and in a natural environment with either models, tower observations or airborne observations, so there is no ability to corroborate the sensitivity of this data to snow in a non-satellite context.

At the same time, fairly similar microwave measurements were studied, and those published studies didn't point to C-band cross-pol being an obvious next step. Prior to this result, the expectation based on the X-band, Ku-band models, tower and airborne observations would be that there is no information in C-band cross-pol in deep mountain snow, or that it was so little information it would be very difficult to extract. The operating principle that leads to the sensitivity that the authors demonstrate is assumed to be depolarization due to scattering, as there is just nothing else it could be. But theory, which has demonstrated usefulness in looking at a whole range of other nearby frequencies based on the work of dozens of researchers over decades has not yet really looked closely at why this works.

I suppose the authors do not fully agree, but maybe they would agree that others probably feel this way? Maybe the introduction could be modified to give some hint as to the divergence in the literature (which the authors have done a nice job discussing in the supplement), and point to that section of the supplement in the intro, and acknowledge that this hasn't yet been demonstrated on the ground? Maybe the "conclusion and outlook" could mention that the authors hope that advanced models such as DMRT + bi-continuous will be able to better tease out the actual mechanism that leads to the demonstrated sensitivity, and that tower-based and airborne studies will be able to corroborate their findings? I very much think those edits would improve the paper!

Reviewer #3 comments

First, I would like to thank the authors for the clarifications made in answering the comments of the 3 reviewers as well as the modifications/additions made in the manuscript. I would like also to thank the two other reviewers for their comments as I share also their concerns on the original version of the manuscript.

Secondly, I am satisfy with the answers to my specific comments. However, I made minor edits to the text to clarify that a C-Band SAR sensor doesn't observe the snow depth but measure the backscattering signal in a given polarisation. The snow depth is derived from an empirical algorithm (equation 2 and 3) using 4000 in-situ observations (or at least 2000) to estimate the parameters α (in m/dB). Also, the authors should underline that the physical explanation of the backscattering ratio decreasing at the end of the winter season is not completely understood. The annotated document is attached.

(Please use the link below to access the annotated document:)

https://drive.google.com/open?id=1Vqlbr_mYRN8Dw-4BBnd4nGgIBCkhhbQX2

2. Reviewer #1 comments

The authors have addressed all my previous comments. As a result, I think the manuscript is almost ready for publication.

My remaining concern is with the new figure S3, which in my view gives a somewhat overly positive view of the correlation between the vh/vv backscatter ratio and snow depth, apparently due to different scales used in the figure axes for snow depth (in S3a the scale for snow depth is from 0 to 2 m, in S3b from 0 to 4 m, and in S3c&d from 0 to 3 m). Unless I am mistaken, this has been done to visually improve the match between vh/vv backscatter and SD in the figure? I think it would be better to either provide uniform scaling across all figures, and accept any possible visual bias that results. If the authors prefer to keep the present format, it would be good to make a note of the scaling in the text (e.g. around line 475 in the supplementary section) and in the figure caption.

We would like to thank you for the effort to review our manuscript and for providing constructive feedback to our work. In the revised manuscript, we have now added a note to the figure caption (Figure 3 in the revised version) to point out the different scales: “**The ranges of the backscatter (dB) and snow depth (m) measurements have been adjusted between sites to improve visualization.**” We believe that a rescaling and better visualization of the temporal dynamics in the backscatter and snow depth measurements is preferred over a uniform scaling with poorer visualization. Biases may be present and better visualized with a uniform scaling, but those are inherent to the comparison of grid-scale against point-scale data and not the focus of this figure.

3. Reviewer #2 comments

Generally the authors have responded well to my concerns raised on the earlier round of reviews. First, I want to say that I find Figure S3 very convincing. Thank you for adding this. Second, the addition of the word

“empirical” I think is excellent, in response to my comment 3 on the previous round. And the additional discussion of empirical literature in the Supplementary section is very helpful.

We would like to thank you for the effort to review our manuscript and for providing constructive feedback to our work. We agree that the addition of Figure 3 (in the revised manuscript), the emphasis on the empirical nature of the algorithm, and the extended literature discussion have significantly improved the paper.

I think there are two things that need to still be improved before publication, however.

First, related to my comment 4 on the previous round of review, you still have to go to the supplement to find that this is an approach based on calibration to a subset of in situ data. The authors’ response to not including “calibration” language in the main text is that they don’t want to include any of the methodology used to compute SWE in the main text in order to avoid confusing readers. I disagree: I think that telling readers that the approach relies on calibrating to a subset of available in situ data, and validating against the rest is both of great value to the readers, and very easy to understand. I think that readers of Nature Communications understand calibration-validation studies, as well as the difference between a measurement based on first principles versus one based on calibration. Please add the language to the main text.

In response to this comment and the editor’s suggestion, we have restructured the manuscript by embedding the supplementary into the main article (see the response to the editor comments). As a result, the full method (including calibration) is now explained in the main article; see “Methods – Sentinel-1 snow depth retrieval algorithm”. Moreover, we added a note earlier in the manuscript that points out the rescaling based on in situ snow depth measurements (P.6, L.152): **“The retrievals of ~weekly snow depth at 1 km² resolution for September 2016 through August 2018 over the Northern Hemisphere mountains are based on the temporal changes in the Sentinel-1 backscatter polarization ratio ($\sigma_{vh}^0/\sigma_{vv}^0$) and scaled to the range of snow depth measurements at in situ sites (Fig. 2), as discussed in the Methods section.”**

Second, it is still not clear in the main article how shocking it is that C-band is sensitive to mountain snow, in the context of decades of tower-based, airborne, and modeling studies on the problem of using active or passive microwave measurements to measure snowpack. The language in lines 64-68 in the current manuscript is: “The use of C-band backscatter for estimating snow depth (or mass, related to depth by the density) has long been swept aside after earlier studies had shown a limited sensitivity [24,25]. However, these studies were mostly surveying shallow snow outside mountain environments and, more importantly, were limited to backscatter measurements in co-polarization.”

Lines 77-82 sort of make it sound like the authors' result is in agreement with previous studies. Lines 64-88 and 77-82 are just not adequate, in my view. From their paper and their response on the previous round of review, the authors have convinced me that previous literature and modeling studies have not shown that C-band cross-pol should not be sensitive to deep snow depth. However, I think they need to go a little further in the main manuscript. In my opinion, I think readers of the main manuscript need to know that in decades of work on remote sensing of snow, no one has actually published results on C-band cross-pol in deep snow and in a natural environment with either models, tower observations or airborne observations, so there is no ability to corroborate the sensitivity of this data to snow in a non-satellite context. At the same time, fairly similar microwave measurements were studied, and those published studies didn’t point to C-band cross-pol being an obvious next step. Prior to this result, the expectation based on the X-band, Ku-band models, tower and airborne observations would be that there is no information in C-band cross-pol in deep mountain snow, or that it was so little information it would be very difficult to extract. The operating principle that leads to the sensitivity that the authors demonstrate is assumed to be de-polarization due to scattering, as there is just

nothing else it could be. But theory, which has demonstrated usefulness in looking at a whole range of other nearby frequencies based on the work of dozens of researchers over decades has not yet really looked closely at why this works.

I suppose the authors do not fully agree, but maybe they would agree that others probably feel this way? Maybe the introduction could be modified to give some hint as to the divergence in the literature (which the authors have done a nice job discussing in the supplement), and point to that section of the supplement in the intro, and acknowledge that this hasn't yet been demonstrated on the ground? Maybe the "conclusion and outlook" could mention that the authors hope that advanced models such as DMRT + bi-continuous will be able to better tease out the actual mechanism that leads to the demonstrated sensitivity, and that tower-based and airborne studies will be able to corroborate their findings? I very much think those edits would improve the paper!

By restructuring the paper, the contrasting results obtained from previous tower-radar experiments are now appropriately described in the "Discussion" section of the main article.

To better highlight (1) the lack of previous experiments with satellite-based C-band cross-pol backscatter measurements and (2) the contradictory results obtained from such tower-based measurements, we rephrased and extended part of the introduction on P.3, L64-72: "The use of C-band backscatter for estimating snow depth (or mass, related to depth by the density) has long been swept aside after **early satellite measurements** had shown a limited sensitivity^{24,25}. However, these studies were mostly surveying shallow snow outside mountain environments and, more importantly, were limited to backscatter measurements in co-polarization. **Cross-polarized backscatter measurements were to-date only investigated at the local scale using tower installations, with strongly contradicting results^{26,27}. Here, we demonstrate the value of including cross-polarized backscatter measurements from C-band satellite to retrieve snow depth in mountainous areas at the large scale.**"

The new results section "Sentinel-1 backscatter signatures over snow" discusses in detail the measurements at different polarizations, with each polarization treated in a separate paragraph. This improves the separation between on the one hand the agreement with previous studies for co-polarization, and on the other hand the absence of prior satellite-based studies for cross-polarization. The revised article now also includes statements (which were previously in the supplementary) that highlight the absence of studies using cross-polarization measurements from satellite imagery, and the contrasting results from tower experiments (e.g., P.11, L.279-282): "**The Sentinel-1 snow depth retrievals are primarily depending on the cross-polarized backscatter measurements.** Temporal variations in σ_{vh}^0 with snow depth evolution have, to our knowledge, not been investigated before with C-band satellite observations. Only few studies deployed tower-mounted radar instruments, with contradictory results."

The main article now includes the discussion of recent advancements in radiative transfer modelling. Besides the text that was transferred from the supplementary, we added (P.12, L.306-309): "**Most of these advances in radiative transfer modelling were thus far focusing on microwave measurements at higher frequencies (e.g., X- and Ku-band). We encourage future modelling efforts to unravel the different mechanisms that cause the demonstrated sensitivity to deep snow at C-band.**" The value of new tower-mounted radar experiments is mentioned on P.12, L.310. Note that our team will initiate such experiments in the coming winter.

4. Reviewer #3 comments

First, I would like to thank the authors for the clarifications made in answering the comments of the 3 reviewers as well as the modifications/additions made in the manuscript. I would like also to thank the two other reviewers for their comments as I share also their concerns on the original version of the manuscript.

Secondly, I am satisfy with the answers to my specific comments. However, I made minor edits to the text to clarify that a C-Band SAR sensor doesn't observe the snow depth but measure the backscattering signal in a given polarisation. The snow depth is derived from an empirical algorithm (equation 2 and 3) using 4000 in-situ observations (or at least 2000) to estimate the parameters α (in m/dB). Also, the authors should underline that the physical explanation of the backscattering ratio decreasing at the end of the winter season is not completely understood. The annotated document is attached.

(Please use the link below to access the annotated document:)

https://drive.google.com/open?id=1VqIbr_mYRN8Dw-4BBnd4nGgIBCkhhbQX2

We would like to thank you for your effort to review our manuscript, for the constructive evaluation of our previous revision, and for the suggested edits.

We agree with the reviewer that our use of the terms “observation” and “measurement” was potentially confusing. In the revised version, we now systematically use “measurement” for backscatter, and “retrieval” for snow depth derived from Sentinel-1. In addition to the edits proposed by the reviewer in the annotated document, we have adopted this consistent terminology throughout the paper. Note that the only place where we did not modify the manuscript according to the suggested edits is the abstract. There, we preferred not to include (the suggested) field-specific terminology such as “SAR” and “C-band”.

By restructuring the manuscript (see the response to the editor comments), the text (P.6, L.138-148) underlining the uncertainties for wet snow conditions towards the end of winter now forms part of the main article instead of the supplementary. To further emphasize that the physical behaviour of the decrease in backscatter at the end of the winter is not completely understood, we modified the text on P.5, L.126 to: “During snow ablation, σ_{vh}^0 decreases considerably, **which we hypothesize is caused by** (i) the absorption and reflection of the signal by wet snow, and (ii) a decreasing amount of snow volume scattering in a shallowing snowpack.”

REVIEWERS' COMMENTS:

Reviewer #2 (Remarks to the Author):

The authors have addressed my concerns. Thank you, I think the paper is much improved as a result. I noticed what I think is a typo: Line 49: I think you mean "observations saturate in deep snow"?

Reviewer #3 (Remarks to the Author):

I have no further concerns and comments. This article will interest a large audience from SAR specialists, radiative transfer modellers and Sentinel-1 data users to hydrologists and water resources managers.

REVIEWERS' COMMENTS:

Reviewer #2 (Remarks to the Author):

The authors have addressed my concerns. Thank you, I think the paper is much improved as a result. I noticed what I think is a typo: Line 49: I think you mean "observations saturate in deep snow"?

Thank you for noticing this typo. We modified the text to: "the observations saturate in deep snow (>0.8 m depth)".

Reviewer #3 (Remarks to the Author):

I have no further concerns and comments. This article will interest a large audience from SAR specialists, radiative transfer modellers and Sentinel-1 data users to hydrologists and water resources managers.